# Structural remodeling of ribosome associated Hsp40-Hsp70 chaperones during co-translational folding

Yan Chen[1,2], Bin Tsai[2], Ningning Li [2] & Ning Gao [2,3,4✉]

Ribosome associated complex (RAC), an obligate heterodimer of HSP40 and HSP70 (Zuo1 and Ssz1 in yeast), is conserved in eukaryotes and functions as co-chaperone for another HSP70 (Ssb1/2 in yeast) to facilitate co-translational folding of nascent polypeptides. Many mechanistic details, such as the coordination of one HSP40 with two HSP70s and the dynamic interplay between RAC-Ssb and growing nascent chains, remain unclear. Here, we report three sets of structures of RAC-containing ribosomal complexes isolated from *Saccharomyces cerevisiae*. Structural analyses indicate that RAC on the nascent-chain-free ribosome is in an autoinhibited conformation, and in the presence of a nascent chain at the peptide tunnel exit (PTE), RAC undergoes large-scale structural remodeling to make Zuo1 J-Domain more accessible to Ssb. Our data also suggest a role of Zuo1 in orienting Ssb-SBD proximal to the PTE for easy capture of the substrate. Altogether, in accordance with previous data, our work suggests a sequence of structural remodeling events for RAC-Ssb during co-translational folding, triggered by the binding and passage of growing nascent chain from one to another.

[1] State Key Laboratory of Membrane Biology, School of Life Science, Tsinghua University, Beijing 100084, China. [2] State Key Laboratory of Membrane Biology, Peking-Tsinghua Joint Center for Life Sciences, School of Life Sciences, Peking University, Beijing 100871, China. [3] National Biomedical Imaging Center, Peking University, Beijing 100871, China. [4] Changping Laboratory, Beijing, China. ✉email: gaon@pku.edu.cn

Co-translational folding of nascent polypeptides is essential for maintaining a functional and healthy proteome in all cells[1–3]. Ribosome-associated chaperones bind to nascent chains after they emerge from the peptide exit tunnel (PET) on the 60S subunit to facilitate their de novo folding[4,5]. There are two types of co-translational chaperones in eukaryotes, the nascent-polypeptide associated complex (NAC) and the ribosome-associated complex (RAC)[4,5]. The composition of RAC as a heterodimer is conserved from yeast to human, with two tightly associated HSP40 (Zuo1 in yeast, ZRF1/MPP11/MIDA1 in mammals) and HSP70 (Ssz1 in yeast, HSP70L1/HSPA14 in mammals) components[6–8]. The function of RAC depends on another HSP70 protein, Ssb1 or Ssb2 in yeast (denoted as Ssb hereafter)[9–11] and HSP70 (HSPA1A/B) in human[12], as a direct binder of the nascent chain to facilitate its co-translational folding. As illustrated in the yeast system, the nascent chain substrates of Ssb are very broad, including ~80% of cytosolic and nuclear proteins, ~80% of nascent mitochondrial proteins and more than 40% of ER-targeted proteins[13,14], manifesting a fundamental role of the RAC-HSP70 system in maintaining basic cellular functions[4,5,15]. Notably, the RAC-Ssb system in yeast was also reported to have additional roles on the ribosome in regulating translation fidelity, namely the stop-codon readthrough, premature termination of polyadenine-containing transcripts and −1 programmed ribosomal frameshifting[16–21].

The co-translational role of the RAC-HSP70 system is well characterized in fungal systems. In fast growing cells of *Saccharomyces cerevisiae*, most RACs are ribosome-bound[22], and the ribosome attachment is mediated by Zuo1[7,9,23]. Absence of any components in the Zuo1-Ssz1-Ssb triad induced similar phenotypes, such as growth defects and high sensitivity to cold, salt and aminoglycosides[7,9–11,24]. Similar to other typical HSP40-HSP70 systems[2], Zuo1 is a J-domain containing HSP40 and acts as a co-chaperone to stimulate the ATP hydrolase activity of Ssb[25,26]. The ATP-hydrolysis on Ssb enables the conformational cycling of Ssb to a high-affinity state (ADP-bound) for stable nascent peptide binding[1,2] until productive folding occurs. Ssb is a typical HSP70, and attaches to the peptide tunnel exit (PTE) region of the ribosome through its SBD[27–29]. The function of Zuo1 in the ATPase stimulation of Ssb requires the RAC as a whole[25]. Unlike canonical HSP70 proteins, Ssz1 binds nucleotides but lacks ATPase activity[25,30–32] and both the nucleotide-binding ability and the C-terminal SBD of Ssz1 are not strictly required for its function[25,30,33]. Most recently, it was demonstrated that both Zuo1 and Ssz1 could be crosslinked with early nascent chain (less than 45 residues) in vivo, and upon the elongation of nascent chain to 50 residues the crosslinks were mostly replaced by Ssb, indicating a substrate relay from RAC to Ssb for the early folding events at the PTE[34].

In addition to this wealth of genetical and biochemical data (reviewed in[15]), mechanistic details of the Zuo1-Ssz1-Ssb system was also under scrutiny of structural studies. Owing to the great flexibility of Zuo1 and RAC either in isolation or on the ribosome[31,35], high-resolution crystal structures are only available for certain fragments of Zuo1 and two partial complexes of RAC, including the C-terminal four-helix bundle domain (4HB)[31,36], the Zuotin homology domain (ZHD)[17], and the N-terminal flexible domain (ND) of Zuo1 in complex with Ssz1[32,34]. Importantly, these two partial Zuo1-Ssz1 complexes revealed extensive interactions of the naturally flexible ND of Zuo1 with the SBD of Ssz1 as a pseudo-substrate, explaining the structural basis of their obligate mutual association throughout the co-translational folding process. On the other hand, low-resolution cryo-EM structures of the 80S-RAC complexes and biochemical characterization have identified the contact sites of RAC near the PTE of the ribosome[17,23,31,37], and more importantly demonstrated the separate binding of the C- and N-terminal parts of Zuo1 to the 40S and 60S subunits, respectively[37]. The CTD (4HB) of Zuo1 interacts with expansion segment 12 (ES12) of the 18S rRNA, which is the peripheral end of the decoding center (DC) helix 44 (h44). The NTD and CTD of Zuo1 is connected by a long and rigid α-helix (middle domain, MD), and therefore this double tethering of two ribosomal subunits poses a constraint to the intersubunit rotation of the ribosome required for peptide elongation[37]. These observations led to an intriguing hypothesis that Zuo1 presents a physical and functional link between the PTE and the DC[17,37].

The multivalent but transient interactions among RAC-Ssb components and between them and the nascent chain, as well as their dynamic association with the ribosome have limited the structural understanding of this process. Many mechanistic questions have remained unanswered. Here, we characterize the structures of RAC-containing ribosomal complexes purified through affinity-tagged Zuo1 or nascent chains. Our results reveal a series of structural rearrangements of RAC upon the engagement with the nascent chain and inform different functional aspects of RAC and Ssb during co-translational folding.

## Results

### Compositional and structural characterization of the endogenous 80S-RAC complexes affinity-purified through tagged Zuo1.
We have previously reported a medium-resolution structure of the in vitro assembled 80S-RAC complex[37]. Because this previous complex was formed with purified empty 80S ribosome, to obtain the native ribosome-RAC complexes, we constructed a yeast strain with the C-terminus of Zuo1 tagged with a 3X-FLAG peptide. Samples were then purified from cell extracts of log-phase yeast cells using anti-FLAG beads and examined by SDS-PAGE and mass spectrometry (MS). Four clear major non-ribosomal bands in the gel were detected, and were identified to be eEF2, Ssa1/2/4, Ssz1 and Zuo1 by MS (Supplementary Fig. 1a; Supplementary Data 1). Ssa1/2/4 are more likely non-specific contaminants, and they stayed in the supernatant after a sucrose-cushion based ultracentrifugation (Supplementary Fig. 1a). Ssb1/2 was also present in the sample, but with a very much lower level compared with RAC (Supplementary Data 1).

Next, cryo-EM single-particle analysis was employed to characterize the endogenous 80S-RAC complexes (Supplementary Fig. 2). Consistent with our previous observations based on purified empty 80S ribosomes[37], in these endogenous 80S-RAC structures, the association of RAC with the ribosome is not dependent on general conformations of the ribosome, i.e., rotated vs non-rotated states (Supplementary Fig. 2; Supplementary Fig. 4a). RAC is extremely dynamic on the ribosome: while the ribosomal part in the density map could reach better than 3-Å resolution, the RAC region remained to be poorly resolved. Therefore, we employed mask-based deep 3D classification to improve the resolution of the PTE-RAC region, which generated three conformational states (A1, A2 and A3) of RAC on the 60S subunit (Supplementary Fig. 2). These states differ in the wobbling positions of RAC on the 60S subunit (Supplementary Fig. 4b). Among them, State A2 could be refined to a global resolution of 3.4 Å (gold-FSC 0.143 standard; Supplementary Table 1), and Zuo1 NTD region could be improved to 5–8 Å with sufficiently resolved secondary structural features (Supplementary Fig. 2, Map 1.2 and Map 1.4; Supplementary Fig. 3).

Surprisingly, although 22.8% of particles contain eEF2, the structures of them do not contain P-site tRNA and lack densities for nascent chain in the PET. In addition, two other factors eIF5A and Stm1 (Fig. 1; Supplementary Fig. 2) were also identified in both the structural and MS data. In fact, Stm1 universally exists in

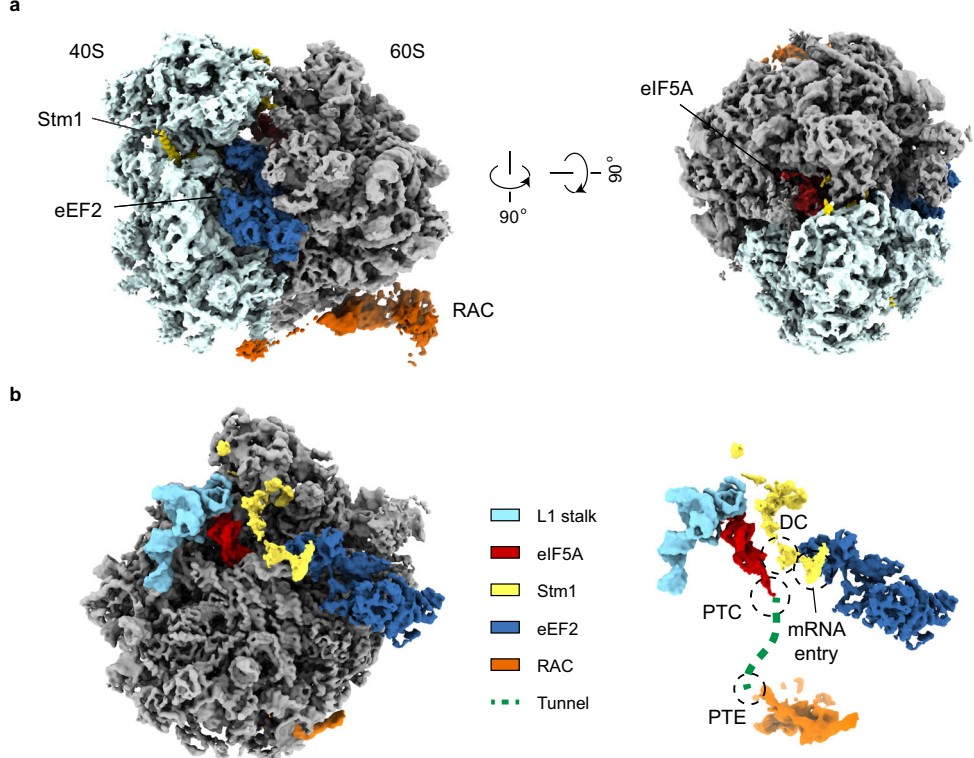

**Fig. 1 Cryo-EM characterization of the endogenous 80S-RAC complexes. a** Overview of the cryo-EM map of the 80S-RAC-eEF2-eIF5A-Stm1 complex, a representative 3D class of the endogenous 80S-RAC complexes. The density map is from Map 1.7 in Supplementary Fig. 2. For purpose of illustration, EM density of RAC is from another map with the RAC region improved by local classification (Map 1.2). The 40S and 60S subunits are colored light cyan and gray, respectively. Ribosome binding factors Stm1, eEF2, eIF5A and RAC are shown in yellow, blue, red and orange, respectively. **b** Subunit-interface view of the endogenous 80S-RAC-eIF5A-eEF2-Stm1 complex. For clarification, only the 60S subunit is shown. The functional centers of the ribosome are highlighted in the right panel. DC, Decoding center; PTC, Peptidyl transferase center; PTE, Peptide tunnel exit. The mRNA entry channel on the 40S subunit is occupied by a helix from Stm1.

all 80S-RAC structures and neither of these structures contains P-site, A/P-site tRNA or the nascent chain. These data indicate that the endogenous 80S-RAC complexes obtained under our experimental conditions are not in canonical elongation cycle, and therefore these Stm1-sequestered structures may not represent genuine snapshots for the role of RAC in co-translational folding.

**Preparation of the substrate-engaged 80S-RAC complexes by nascent chains.** Since the endogenous 80S-RAC complexes purified through tagged Zuo1 did not contain P-site tRNA and nascent polypeptide chain, to obtain a sample more relevant to co-translational folding, we sought to design a ribosome-nascent chain complex (RNC) that bears a native substrate of the RAC-Ssb system. We constructed a plasmid encoding the N-terminal sequence of Pmt1, which is one of the reported Ssb substrates[14], and introduced several translation stalling motifs to the C-terminus of Pmt1 peptide to synchronize nascent chains (Supplementary Fig. 5a). To minimize the stalling-induced degradation of nascent chain and mRNA by ribosome-associated quality control (RQC) system[38,39], these constructs were expressed in a mutant yeast strain lacking Dom34 (Δ*dom*), a factor required for splitting stalled ribosomes in the RQC process[40]. Based on the yield of the RNCs, the triple-proline motif (PPP) was finally chosen for the preparation of the RNC complexes from Δ*dom* cells (Supplementary Fig. 5b, c). Also, to maximize the occupancy of RAC on the RNCs, recombinant RAC proteins were added to the crude ribosomal pellets during RNC purification (see methods for details), and RAC was thus co-purified as a component of RNCs. As indicated by SDS-PAGE

and MS analyses (Supplementary Fig. 1b; Supplementary Data 1), the purified RNC complexes are indeed enriched in the factors of co-translational folding, including Zuo1, Ssz1 and Ssb1/2. Additional factors, such as eEF2 and other ribosome-binding proteins are also present in the sample (Supplementary Fig. 1b). On the basis of these purified RNC-RAC complexes, another sample (RNC-RAC-Ssb1) was derived by supplementing purified Ssb1 in the presence of ATP. These two samples were then characterized by cryo-EM (Supplementary Figs. 6, 7, 9, 10).

**Structural characterization of the RNC-RAC-Ssb1 and RNC-RAC datasets.** The RNC-RAC-Ssb1 dataset was processed similarly as we did for the dataset of the endogenous 80S-RAC complexes (Supplementary Fig. 6). Although ribosome oligomers, including dimers and trimers, were observed during 2D and 3D classification, most of the RNC-RAC particles (89%) are monosomes. Unlike the inactive endogenous 80S-RAC complexes, structures of these RNC-RAC-Ssb1 classes are indeed snapshots of translating ribosomes in the elongation cycle. The ribosomes in these structures are all in differently rotated positions with both the A-site and P-site occupied by tRNAs, and the density of nascent chain extending from the A- or A/P-site tRNA is clearly discernible in the PET (Supplementary Fig. 8a, b). Through focused classification, we also identified three major states (B1, B2 and B3) with more continuous densities of Zuo1 (Supplementary Fig. 6; Supplementary Fig. 4c), and the best of the three (B2) could be refined to a global resolution of 2.9 Å (Supplementary Fig. 6, Map 2.2; Supplementary Fig. 7a, c; Supplementary Table 1). The ZHD and JD region of Zuo1 in this structure could

be further improved to 4-Å resolution, (Supplementary Fig. 6, Map 2.4; Supplementary Fig. 7b), with secondary structures matching with the crystal structure of Zuo1 ZHD (PDB 5DJE)[17] and a homology model of Zuo1 JD derived from the crystal structure of DnaJ[41]. Very intriguing, during deep 3D classification, a small fraction of particles constituted an unexpected Stm1-containing structure, which bears nascent chain in the PET but no tRNA (Supplementary Fig. 6, Map 2.8; Supplementary Fig. 8c, d). Overall, a major difference from the endogenous 80S-RAC dataset is that there is no solid density for Ssz1 in all these RNC-RAC-Ssb1 structures, indicating that Ssz1 is highly flexible in this sample.

The RNC-RAC dataset was processed to focus on conformational sorting of RAC (Supplementary Fig. 9). Unlike the RNC-RAC-Ssb1 dataset, solid density for both Zuo1 and Ssz1 could be found during 3D classification. Four major conformational groups (C1-C4) representing distinct positions of Ssz1 were identified. The C1 group is generally similar to the one from the endogenous 80S-RAC dataset (Supplementary Fig. 2, Map 1.4; Supplementary Fig. 9, Map 3.1), while Ssz1 in the C2 and C3 groups has dramatically changed its position and flipped to the PTE side of Zuo1(Supplementary Fig. 9, Map 3.2 and Map 3.3). The best class C2 could be refined to a global resolution of 3.0 Å (Supplementary Figs. 9, Map 3.5; Supplementary Fig. 10a, c; Supplementary Table 1). After local refinement, the region of RAC has well clearly separated secondary structures (Supplementary Fig. 9, Map 3.6; Supplementary Fig. 10b), and could fit nearly full-length proteins for both Zuo1-NTD and Ssz1 (Supplementary Movie 1). In contrast, in the remaining classes (C4), whereas Zuo1-NTD is fairly resolved no solid density of Ssz1 was found except those highly fragmented pieces near the PTE.

**Orientation of Zuo1 JD relative to the ribosome**. Based on these improved maps of Zuo1-NTD, a complete pseudo-atomic model for the ZHD and JD could be derived, which is a relatively single rigid piece. The JDs of typical J-domain-containing HSP40 proteins consist of four helices, the second of which has a conserved HPD motif essential for the stimulation of the ATPase activity of HSP70 proteins[42]. A single mutation of H128Q in Zuo1-HPD was able to completely abolish the in vivo function of Ssb without impact on the binding of RAC to the ribosome[10]. Based on the low-resolution cryo-EM structure of 80S-RAC[37], a rough position of Zuo1 ZHD on the ribosome was previously determined[17]. Our structure thus enabled a more accurate positioning of both ZHD and JD on the 60S subunit, and more importantly, allowed the understanding of how Zuo1 JD is orientated relative to the PTE (Fig. 2a, b). Zuo1 JD sits on the solvent-exposed side of ZHD, free of any ribosomal contact, and the HPD motif is in the distal end of Zuo1, being the farthest point from the PTE.

**Dynamic interactions of Zuo1 with the 60S subunit at the PTE**. Our structures indicate that RAC is highly dynamic at the PTE and displays both translational and rotational of movement relative to the 60S subunit (Supplementary Fig. 4). The pseudo-atomic models of Zuo1-NTD in different conformational states of the RNC-RAC-Ssb1 dataset allowed the analysis of the structural basis of the RAC mobility on the 60S subunit. Superimposition of models from States B1 to B3 suggests that the JD-ZHD wobbles around the axis of H24-eL31-H101 (Fig. 2b), resulting in the swinging of the whole RAC on the 60S subunit (Supplementary Fig. 4c). The HPD motif could move up to 20 Å between State B1 and B3 (Fig. 2c).

In these states, the N-terminal region of helix III (α3) in Zuo1 ZHD keeps anchored with H24[17] through two highly conserved arginine residues R247 and R251 (Fig. 2d, g). At the C-terminal

region of helix III, a few charged residues of eL31 on the ribosomal surface appear to generate a platform for dynamic interactions with helix II or III of the ZHD (α2/α3). At this adjustable interface, eL31 could accommodate the ZHD in different wobbling positions (Fig. 2e, h). The MDs from different states roughly converge at the same site near H101 (Fig. 2f, i), suggesting a second anchor point for Zuo1 on the 60S subunit. Two basic amino acids (K301 and R304) in the MD play a major role in interacting with H101, and the interactions appear to be in different strength, from strong to weak in State B1 to B3 based on the distances between the MD and H101 in these states (Fig. 2i).

Therefore, our structural data suggest a model for the dynamic interactions of RAC with the 60S subunit, in which the wobbling of RAC is enabled by two anchor points, H101 and H24. Consequently, to achieve simultaneous fixed binding of H24 and H101, the helix α3 of Zuo1 ZHD has been deformed to different extents near the PTE (Fig. 2c). This model is supported by ours and previous genetic data. Double mutation of R247 and R251 into alanines to disrupt H24 interaction completely phenocopied the knockout strain of Δzuo (Supplementary Fig. 11)[43]. At the H101 interface, simultaneous mutation of the three basic residues (K301A-R304A-R309A) also could not fully rescue the growth defects of Δzuo cells. We also tested the less rigid interface of Zuo1 with eL31 by generating a mutant strain harboring a double mutation of eL31, R79A and E81A, which are the two closest residues to the ZHD (Fig. 2e, h), on both genomic copies of eL31 genes (RPL31A/B). This strain displayed no growth difference from the WT strain (Supplementary Fig. 11), in consistent with previous data that eL31 is not essential for the binding of RAC to the 60S subunit[23].

Altogether, our structures demonstrate that RAC employs both rigid and versatile ribosomal contacts to maintain a dynamic association with the 60S subunit, and the rigid rRNA contacts are more important for its association.

**The 4HB domain of Zuo1 modulates ES12-h44 conformation**. Previously, we reported the interaction of the C-terminal 4HB domain of Zuo1 with ES12 (helix 44 of the 18S rRNA) in the low-resolution structure of the 80S-RAC complex[37], and this contact site was further biochemically examined[17,44]. Based on the RNC-RAC-Ssb1 dataset, we obtained two density maps, representing two conformational states of h44, the 4HB-bound and 4HB-free states (Supplementary Fig. 6, Map 2.5 and Map 2.7), and the region of 4HB-ES12 in the final map could be improved to 6 Å by focused refinement (Supplementary Fig. 6, Map 2.6; Supplementary Fig. 7d-f). This much-improved map allowed us to accurately fit the crystal structure of the 4HB[31,36] to analyze the interface (Fig. 3a, b).

Specifically, the C-terminal end of Zuo1 MD (residues S335 to A350) inserts into the major groove of ES12 (Fig. 3b, c). This helix contains as many as six lysine residues (Supplementary Fig. 12), and based on our model, five of them (K337/341/342/344/348) are close to the phosphate backbone of ES12 (Fig. 3c). Notably, there is no physical contact between the main body of Zuo1 4HB and ES12, except that K352 and R356 in the first helix of 4HB (the extension of the helical MD) are proximal to the phosphates of ES12. This specific embedding of a highly positively charged α-helix in the major groove could tightly lock the 4HB on ES12. Compared to the 4HB-free state, ES12 is seen to have a large rotation (~15°), with up to 13-Å displacement at the tip (Fig. 3d).

It was previously shown that several single lysine (K341A, K342A, K344 A and K348A) or a triple-lysine (K341A-K342A-K344 A) mutations in this ES12-interacting helix of Zuo1 all reduced the binding of Zuo1 to the ribosome in vivo, but had

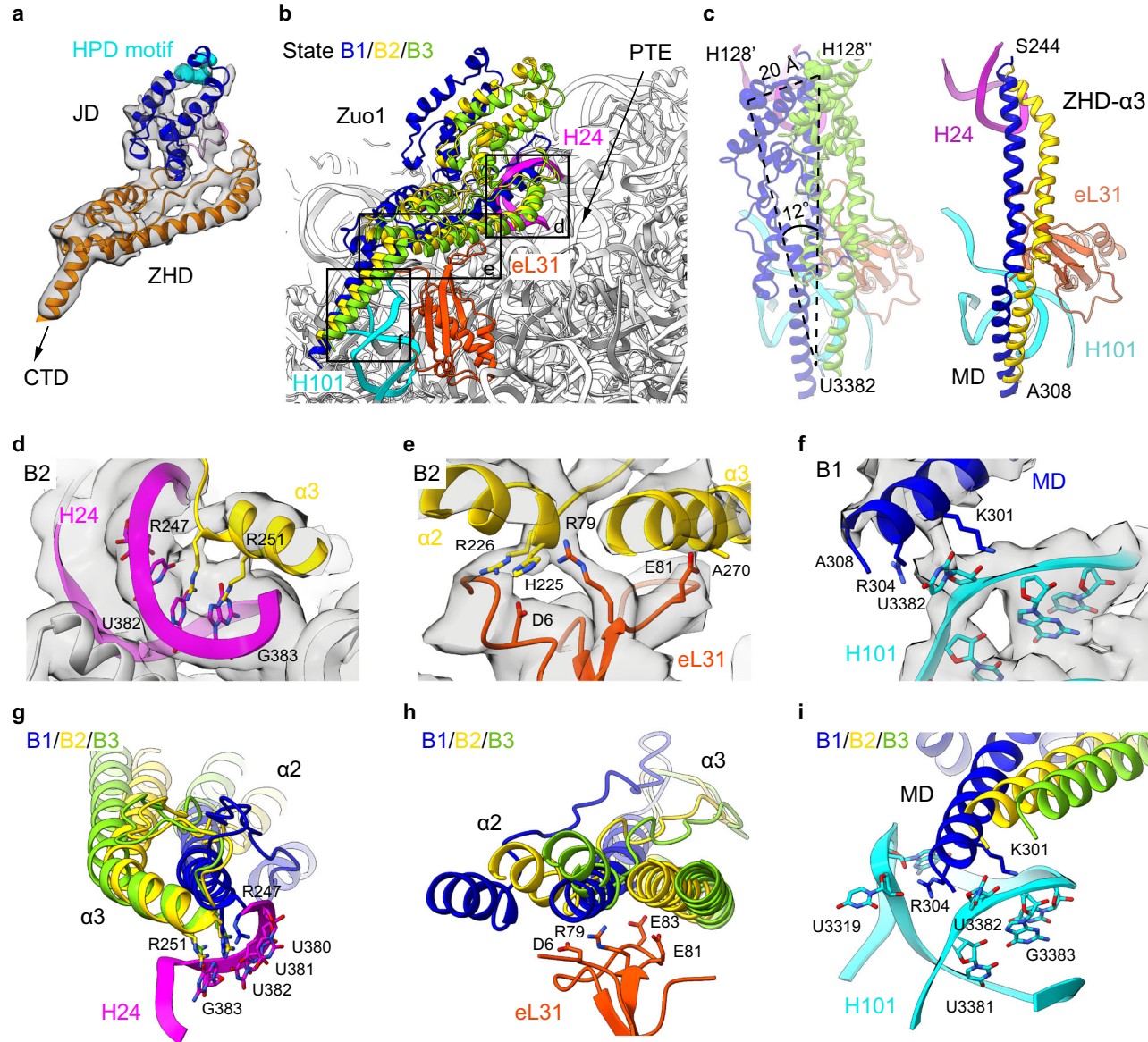

**Fig. 2 Dynamic interaction of Zuo1 with the 60S subunit at the peptide exit. a** Segmented density map of Zuo1 ZHD and JD (from the RNC-RAC-Ssb1 dataset), superimposed with the atomic model. JD and ZHD are colored blue and orange, respectively. The HPD motif is highlighted in cyan spheres. The density map is from a focused refinement (Map 2.4 in Supplementary Fig. 6). **b** Magnified view of the 60S subunit in the RAC binding region. Zuo1 models of State B1, B2 and B3 are colored blue, gold and green, respectively. H101, eL31 and H24 are colored cyan, orangered and magenta, respectively. **c**, Comparison of the models of Zuo1 on the 60S subunit. The HPD motif in Zuo1 JD displays up to 12° rotation and 20-Å shift with respect to U3382 of the 25S rRNA (left). The flexibility of ZHD-α3 is the structural basis of the dynamics of RAC on the 60S subunit (right). **d**–**f** Contact points of Zuo1 with the 60S subunit. Detailed views of interactions between the N-terminus of ZHD α3 and H24 (**d**) in State B2, between ZHD α2/α3 and eL31 (**e**) in State B2, and between MD and H101 (**f**) in State B1. The atomic models of Zuo1 are superimposed with EM density from State B2 (**d**, **e**) or State B1 (**f**). Selected residues of Zuo1, eL31 and 25S rRNA are shown in stick models and labelled. The density maps are from Map 2.2 and Map 2.1 in Supplementary Fig. 6. **g**–**i**, Structural comparison of the models of Zuo1 (State B1-B3) with respect to the three ribosomal contact points, H24 (**g**), eL31 (**h**) and H101 (**i**).

marginal effect on cell growth[17]. We therefore examined whether a combination of more mutations in this lysine-enriched helix affects the growth of yeast cells under low temperature or paromomycin-treated conditions. As shown, mutations of four lysine residues simultaneously (4A) in this helix led to apparent growth defects in both conditions (Supplementary Fig. 11). In addition, a double mutation of K352A-R356A, which are two newly identified residues at the interface, resulted in an even higher sensitivity to these two stresses, compared with the 4A mutant (Supplementary Fig. 11). Thus, together with previous genetic data[9,17,37], a conclusion is that multiple basic residues of Zuo1-CTD contribute

to its binding to ES12, and the interaction between Zuo1-CTD and ES12 is required for the optimal function of RAC in vivo.

Our structures show that Zuo1 4HB deforms ES12 in a specific way that would pose a tension to h44. Since the upper portion of h44 in the 4HB-bound state appears to be in the same conformation as in regular translating ribosomes (Fig. 3d), the effect of 4HB-ES12 interaction on the decoding center could be very subtle. Zuo1 could potentially affect the conformational dynamics of the DC through the terminal end of h44, to kinetically modulate the interactions of translation factors with the decoding center. Therefore, in addition to an apparent role of

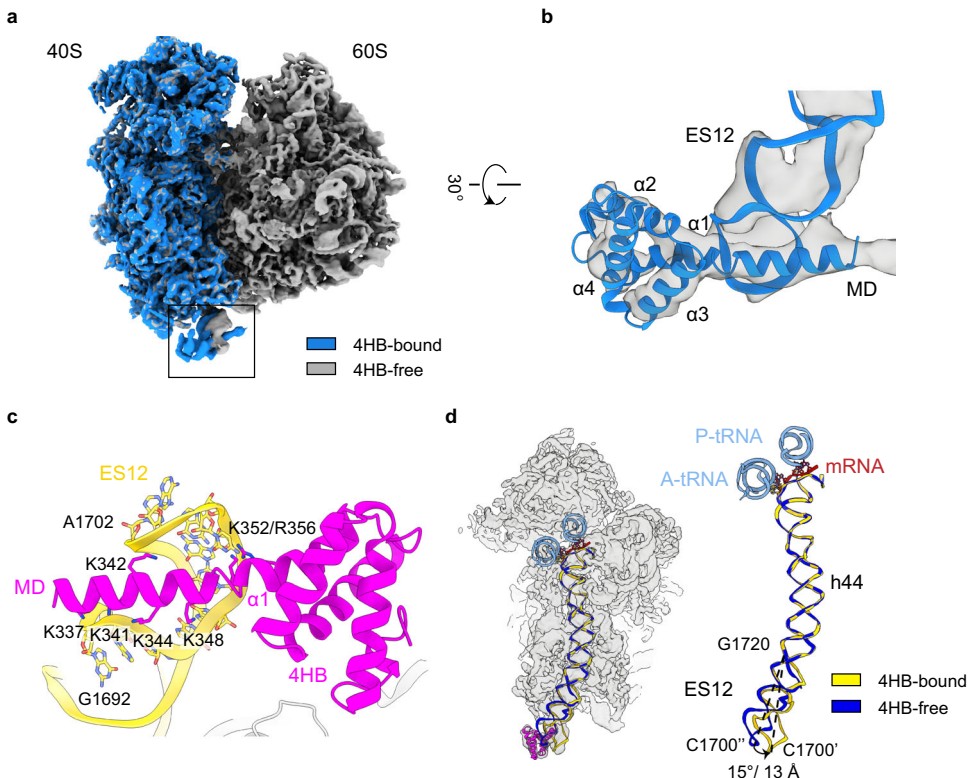

**Fig. 3 Conformational modulation of h44 by RAC 4HB. a** Cryo-EM maps of 4HB-bound (blue) and 4HB-free (gray) RNCs. The density maps of the two RNCs are from Map 2.5 and Map 2.7 in Supplementary Fig. 6. The alignment was done with the 60S subunit as reference. **b** Magnified view of the boxed region in **a**, highlighting the fitting of Zuo1 4HB into the cryo-EM density (Map 2.6). **c** Molecular model of interactions between basic residues of MD-4HB and ES12. MD-4HB and ES12 are colored magenta and gold, respectively. **d** Structural modulation of h44 by RAC-ES12 interaction. Cryo-EM map of the 40S subunit in 4HB-bound state (Map 2.5), superimposed with the atomic model of the MD-4HB and the models of h44 in the two states. The models of h44 from 4HB-bound and 4HB-free RNC are colored gold and blue, respectively. The mRNA and tRNAs are shown in red and light blue, respectively.

Zuo1 in modulating the intersubunit rotation required for tRNA translocation on the ribosome, Zuo1 could potentially regulate the conformational dynamics of the decoding center to affect other steps of translation.

**Interaction of Ssz1 with the ribosome at the PTE.** Unlike the RNC-RAC-Ssb1 dataset, which offers no structural information for Ssz1, three meta-stable conformational states of Ssz1 were found from the RNC-RAC dataset (Supplementary Fig. 9). The best resolved class C2 enabled us to generate a reliable model for both Zuo1 and Ssz1 (Fig. 4, Supplementary Movie 1), based on the crystal structure of $Ct$Ssz1-ND$_{Zuo1}$ complex[34] and the AlphaFold2 predicted model of $Sc$Ssz1[45]. Except for a disordered region of Zuo1 (residues 50–70) and a highly flexible β-hairpin of Ssz1 (residues 458–488), the cryo-EM map has well resolved densities for all rest parts of Zuo1-NTD and Ssz1. (Fig. 4a, b). The Zuo1 ND all the way up to the first N-terminal residue, which tightly interacts with both the NBD and SBD of Ssz1[34], is also discernible in the map (Fig. 4b, c).

In this specific conformation, in addition to the previously characterized interactions of Zuo1 ND with Ssz1, two new inter-molecular contact sites were identified. The first involves the first helix of Zuo1 ND (residues 33–44), which is sandwiched between the HPD-containing helix of Zuo1 JD and the IA subdomain of Ssz1 NBD (Fig. 4c, Supplementary Movie 1). The second is between the surface of Ssz1 SBD (R521, K534) with the N-terminal sequence of Zuo1 JD (E83, D89), through polar interactions (Fig. 4c). Very importantly, this conformation of Ssz1 is stabilized by a previously unrecognized interaction of a sequence on the IIB subdomain of Ssz1 NBD with uL24 and

the H7-loop of the 25S rRNA (Fig. 4d). This interface was resolved at side-chain resolution. Specifically, H293 of Ssz1 stacks with the base of G85 in H7-loop; S282 and D284 of Ssz1 establish charged interactions with H110 of uL24 (Fig. 4d; Supplementary Movie 1). Upon this interaction, the H7-loop is seen to undergo a conformation change, being pushed away from the H7:uL24 interface (Fig. 4d).

**Remodeling of RAC on the ribosome when engaging with the nascent chain.** These metastable structures of the ribosome-RAC complexes from the three datasets allow interesting side-by-side comparisons. Firstly, in the density map of the State C1 from the RNC-RAC dataset, with the fitting of Zuo1 ZHD and JD, the density of Ssz1 could be identified (Fig. 5a, e). The rigid-body fitting of Ssz1 model indicates that Ssz1 interacts with Zuo1 mainly through its SBD, with the NBD free of ribosomal contact (Fig. 5e). This conformation closely resembles that of RAC in the endogenous 80S-RAC complex (Supplementary Fig. 2, Map 1.4; Supplementary Fig. 9, Map 3.1). Since the structure of the endogenous 80S-RAC complex does not have a nascent chain, this C1-type conformation likely represents a resting state for RAC on the ribosome that is free of nascent-chain interaction. Secondly, in the State C2 of the RNC-RAC complex, Ssz1 undergoes a very large rotation to flip to the opposite side of Zuo1. In this conformation, Ssz1 sits exactly above the PTE (Fig. 4; Fig. 5b; Supplementary Movie 1), placing its SBD close to the exit. Thirdly, in the relatively less resolved State C3 of the RNC-RAC complex, although the density of Ssz1 is not complete, Ssz1 stays on the same side of Zuo1 as the State C2 but the SBD detaches from Zuo1 JD and increases its distance from the PTE.

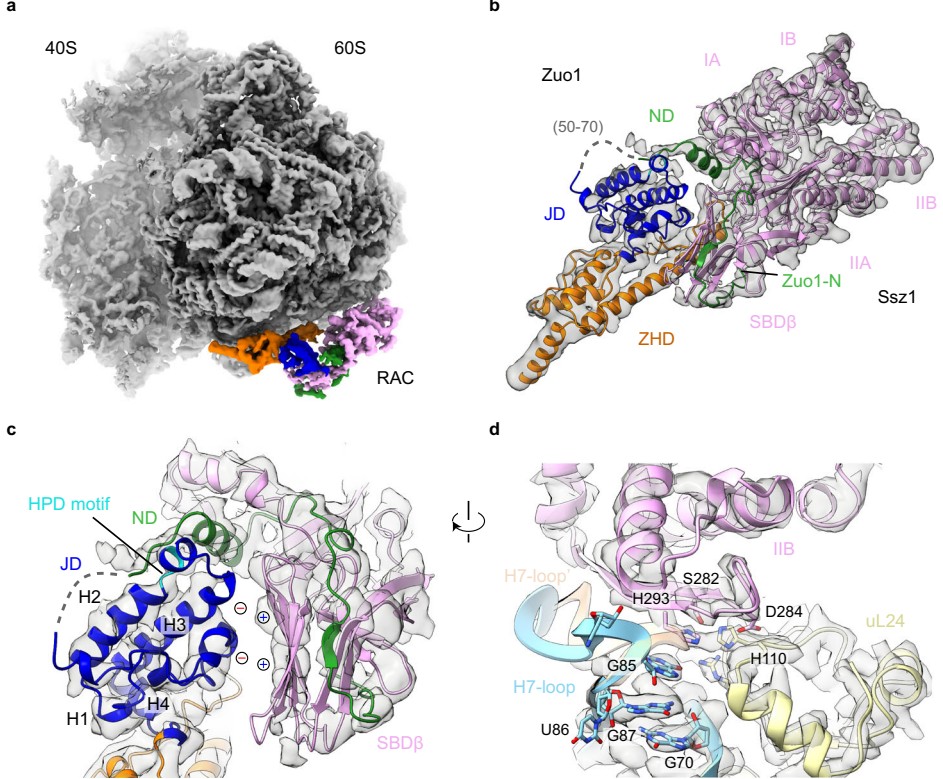

**Fig. 4 Structure of the RNC-RAC complex with nearly full density of RAC. a** Overall structure of the State C2 from the RNC-RAC dataset. The density map is from Map 3.5 in Supplementary Fig. 9. The ribosome is colored gray, and Zuo1 ND, JD, ZHD and Ssz1 are shown in green, blue, orange and pink, respectively. **b** Segmented density map of RAC, superimposed with the models of Zuo1 and Ssz1. The subdomains of both proteins are labelled, and the flexible sequence of Zuo1 (50–70) is indicated in a gray dash-line. The density map is from Map 3.6 in Supplementary Fig. 9. **c** Magnified view of the intermolecular interface between Zuo1 JD and Ssz1 SBD. Zuo1 ND partially blocks the HPD motif (highlighted in cyan) of Zuo1 JD. The polar interface between the surface of Zuo1 JD helices and Ssz1 SBD is highlighted by circled "+" and "-" symbols. **d** Ribosomal contacts of Ssz1 at the PTE. Ssz1 interacts with uL24 and H7-loop, a component of the 25S rRNA, through its subdomain NBD-IIB. Ssz1 and uL24 are shown in pink and light yellow, respectively. H7-loop, with or without Ssz1 interactions, is colored light blue and apricot, respectively. Selected residues of Ssz1, uL24 and H7-loop are shown in stick models and labelled.

Lastly, in sharp contrast to these conformations, the structures from the sample supplemented with Ssb1-ATP lack stable density for Ssz1 and Zuo1 ND, indicating that they are completely flexible (Fig. 5d). Since we included ATP in the sample preparation, the structures from the RNC-RAC-Ssb1 dataset should reflect the states after the ATP-hydrolysis and Ssb1 in the sample is in ADP-bound conformation.

Correlating the compositions of these three samples with their structural features, it suggests that the nascent chain plays an important role in mobilizing Ssz1 on the ribosome. In all the nascent chain-free 80S-RAC structures (Supplementary Fig. 2), the densities of Ssz1 are on distant side of Zuo1 from the PTE. In the presence of a nascent chain substrate, while a small fraction of RNC-RAC particles (C1) remain in the resting state, most of them are associated with Ssz1 densities in the PTE side of Zuo1. Both Zuo1 and Ssz1 could be crosslinked to the nascent chain in vivo[34], hinting that this remodeling of Ssz1 likely depends on the interaction of the emerging nascent chain with Zuo1 and/or Ssz1.

Notably, transition from C1 to C2 would require substantial conformational changes of both Zuo1 and Ssz1 (Fig. 5e–h). As Zuo1-NTD takes a rotation around the H24-eL31-H101 axis in a direction away from the PTE with a 20-Å displacement on the HPD motif (Fig. 5h), Ssz1 undergoes a nearly 180°-rotation roughly round the HPD motif (Fig. 5g). Given the relative size of these two HSP70 proteins in the limited space of the PTE, upon the addition of Ssb1-ATP, Ssb would compete with Ssz1 for the

general location of the PTE. Indeed, Ssb SBD was reported to be capable of interacting with the ribosome[27,28], and its SBDβ could be crosslinked with uL24 in vivo[29]. Therefore, successive extensive structural remodeling of RAC ought to take place, first upon Ssb-ATP binding and then after ATP-hydrolysis of Ssb. This explains why no solid densities were found for Ssz1 in the structures of the RNC-RAC-Ssb1 dataset.

**Mechanism of RAC in promoting the chaperone function of Ssb.** Based on the structural studies of the J-domain containing HSP40 and HSP70 proteins[46,47], the interaction between JD and HSP70 is highly conserved. Therefore, the initial binding position of Ssb-ATP relatively to the JD of Zuo1 could be determined by structural alignment to the previous structure of the DnaK-DnaJ complex[46]. When the structure of Ssb1 in ATP-bound state[27] is overlaid on Zuo1 JD based on the conserved HSP40:HSP70 interface, there is no steric clash between the ribosome and Ssb1 (Fig. 6a), implying that this conformation of Ssb1 should reflect the state of initial docking of Ssb1 on the ribosome. However, superimposition of this model on the structures of the endogenous 80S-RAC complex and the State C1 of the RNC-RAC complex would result in large steric clash between Ssz1 SBD and Ssb1 NBD (Fig. 6b), indicating that this conformation of RAC (C1-type) is incompetent to stimulate the ATPase activity of Ssb. Therefore, it again suggests that the C1-type state likely reflects the resting conformation of RAC, in which the HPD motif of Zuo1 JD is covered by Ssz1 SBD (Figs. 5a, e and 6b).

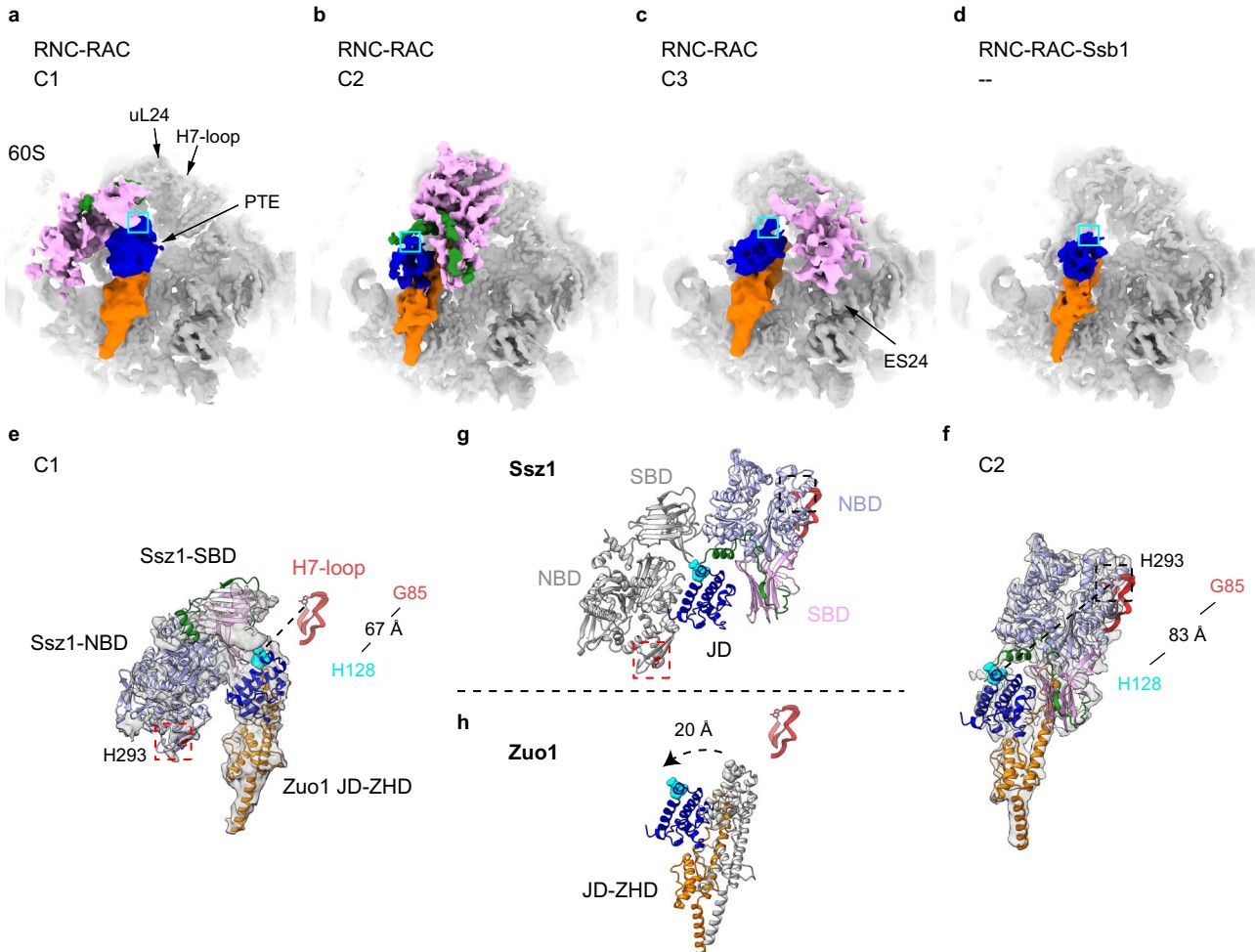

**Fig. 5 Structural remodeling of RAC when engaging with the nascent chain. a–c** Magnified views at the PTE region of State C1-C3 from the RNC-RAC datasets (Map 3.1, Map 3.5 and Map 3.3 in Supplementary Fig. 9). The ribosome is colored gray, and Zuo1 ND, JD, ZHD and Ssz1 are shown in green, blue, orange and pink, respectively. The location of the HPD motif is labelled by a cyan box, and the PTE, uL24, H7-loop and ES24 is labeled. **d** Magnified view at the PTE region of the structure from the RNC-RAC-Ssb1 dataset (Map 2.2 in Supplementary Fig. 6). **e–h** Structural transition from State C1 (**a**) to C2 (**b**). Nearly 180°-rotation of Ssz1 roughly round the HPD motif (Zuo1 JD as reference for alignment) (**g**) and rotation of Zuo1-NTD away from the PTE (**h**). Ssz1 NBD and SBD are colored lavender and pink, respectively. Models of RAC in State C1 before the transition in **g**–**h** are colored gray. The ribosome-interacting region in Ssz1 NBD-IIB is labelled by dash-line box in red (C1) or black (C2). H7-loop of the 25S rRNA is also shown as a landmark for comparison, with the nucleotide G85 in stick model. Upon the transition, the distance between G85 and the H128 of the HPD motif increases from 67 to 83 Å.

In the C2 and C3 structures from the RNC-RAC dataset, Ssz1 now has moved to the PTE side of Zuo1. Although direct alignment of the model of Ssb-ATP would still generate steric clash with Ssz1 (Fig. 6b), the Zuo1 JD interface for Ssb-ATP becomes more exposed (Fig. 5b, c). Thus, it appears that one advantage of RAC remodeling upon nascent chain engaging is to allow Ssz1 to sample more conformations to fully expose Zuo1 JD. In support of this idea, Zuo1 JD is indeed fully accessible in the structures of the RNC-RAC-Ssb1 dataset (Figs. 5d and 6a).

Another observation from the structural alignment is that the SBDβ of Ssb1-ATP is placed in a position close to the PTE (Fig. 6c). This positioning of Ssb1 SBD is of physiological significance, which implies that upon binding to the JD of Zuo1, Ssb1 SBD is brought close to the PTE for easy capture of the substrate. In the aligned model of Ssb1-ATP, the substrate binding pocket of Ssb1 SBD is facing the PTE, with the substrate fragment (superimposed based on a substrate-bound DnaK structure)[48] right at the tunnel exit (Fig. 6c). This possibly explains the previous observations that most Ssb-binding motifs are short linear hydrophobic elements and Ssb binds them when the nascent chain extends 10–15 residues from the tunnel exit[13,14,34].

Moreover, the map of the C1 structure reveals a direct contact between the nascent chain density with the N-terminus of the α3 helix of Zuo1 ZHD (Fig. 6d), indicating that Zuo1 is the first binder of the nascent chain among the three RAC-Ssb components. Nascent chain beyond this point is untraceable. This contact site of Zuo1 is exactly a hydrophobic surface (Fig. 6e), comprising of H249, Y252, I253, which is likely responsible for the previously reported in vivo crosslinking between Zuo1 and nascent chain[34]. Intriguingly, the C2 conformation of Ssz1 would place the very-N terminus of Zuo1 ND next to the nascent chain contact site of Zuo1 ZHD-α3 (Fig. 6e), highlighting a possible competition between Zuo1 ND and nascent chain for the common substrate binding site on Ssz1 SBD[34]. In the map of the C2 structure, due to the tilting of Zuo1 ZHD (Fig. 5g), the direct connection between ZHD-α3 and the nascent chain density is gone (Fig. 6f). Instead, a strong density piece, bout 3-4 residues long, appears above the N-terminus of ZHD-α3, very close to the very-N terminus of Zuo1 ND (Fig. 6f). This could be a fragment of the nascent chain, but other assignment remains possible because it is within the distance of the disordered β-hairpin loop of Ssz1 SBD (Supplementary

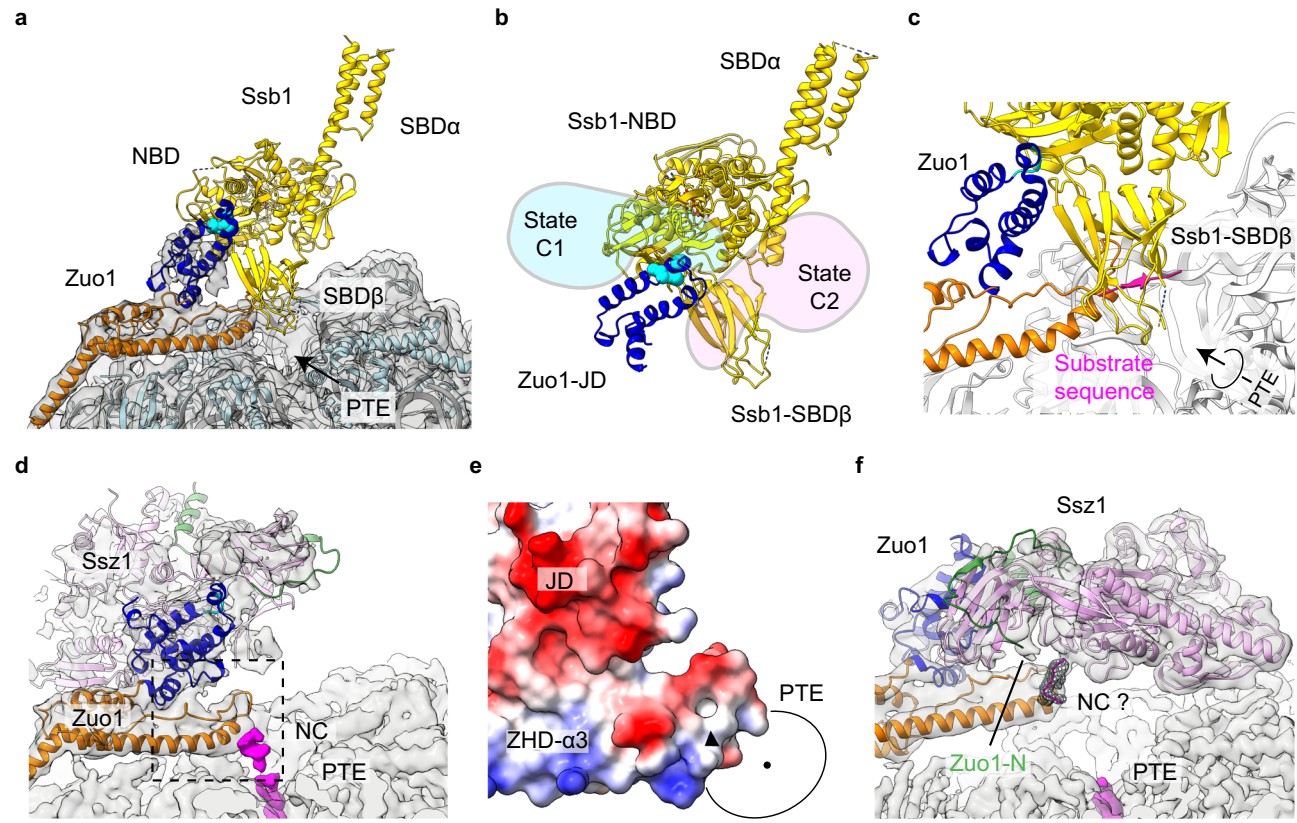

**Fig. 6 Docking of Ssb-ATP on the 80S structure of the RNC-RAC-Ssb1 dataset. a** Superimposition of the atomic model of Ssb1-ATP (5TKY) onto the RNC-RAC-Ssb1 structure (Map 2.2 in Supplementary Fig. 6), based on the structural alignment to JD$_{DnaJ}$-DnaK-ATP complex (PDB 5NRO). Ssb1 is colored gold. There is no steric clash between Ssb1 and the 60S subunit. **b** Steric clash between Ssb1-ATP and Ssz1 in State C1 and C2 from the RNC-RAC dataset. **c** Magnified view of the PTE region in **a**. A putative substrate (magenta) of Ssb1 SBD was derived from the crystal structure of DnaK-substrate complex (PDB 4R5I). The arrow indicates the direction of nascent chain elongation. **d** Magnified view of the PTE region in the structure of State C1 from the RNC-RAC dataset (Map 3.1 in Supplementary Fig. 9). Nascent chain is shown in magenta. **e** Boxed region in **d** shown in electrostatic surface. The negatively and positively charged surface is shown in red and blue, and the hydrophobic surface in white. The black triangle indicates the contact site of the nascent chain in State C1 (**d**). **f** Magnified view of the PTE region in the structure of State C2 from the RNC-RAC dataset (Map 3.5 in Supplementary Fig. 9). Nascent chain in the peptide tunnel is shown in magenta. A strong density piece, that might be a fragment of the nascent chain, is highlighted in mesh representation.

Movie 1). Therefore, the conformations of RAC in the structures of the RNC-RAC dataset might reflect different binding patterns of nascent chain with Zuo1 and Ssz1.

In sum, our structural analyses demonstrate that the interaction of nascent chain with RAC regulates the dynamics of RAC and serves as a switch to control the accessibility of Zuo1 JD for Ssb-ATP recruitment. In addition, Zuo1 might have a molecular role in orienting Ssb on the ribosome to promote the substrate capture of Ssb and to facilitate the establishment of ribosome-Ssb interaction. This is consistent with the current notion that the functional interplay between RAC and Ssb is beyond the Zuo1-mediated ATPase stimulation of Ssb. RAC is required for the optimal interaction of Ssb SBD with the ribosome in vivo[27], and mutant Ssb with abolished autonomous ribosome-binding ability critically depends on RAC for substrate interaction[28].

## Discussion

**Model for the actions of RAC on the ribosome**. In this work, we characterized the structures of RAC-containing ribosomal complexes prepared from fast growing yeast cells. The samples were obtained from two purification strategies, with an affinity tag on the C-terminus of Zuo1 or the N-terminus of engineered nascent chain. While the endogenous 80S-RAC complexes sampled through Zuo1 were surprisingly found to be inactive and contain

no nascent chain, the structures of the RNC-RAC complexes, with both tRNAs and nascent chains, are authentic snapshots of RAC during co-translational folding. Although the RNC was prepared with a highly engineered nascent chain, our analyses indicate that these structures could at least structurally reflect distinct functional stages of RAC on the ribosome. Through integration of recent data[15,17,27–29,32,34], and in light of the proposed substrate-relay model[34], our results reveal a structural framework for understanding the interplay between RAC and Ssb, involving sequential conformational changes of Zuo1, Ssz1 and Ssb1, during co-translational folding (Fig. 7).

First, upon the initial binding of RAC to the ribosome, Zuo1 and Ssz1 may have a relatively rigid interface. In this state, the HPD motif of Zuo1 JD is covered by Ssz1 SBD, and the short nascent chain (~5 residues from the PTE) starts to establish its first contact with the hydrophobic patch on the ZHD of Zuo1 (Fig. 7a, b). Next, the elongation of nascent chain triggers extensive conformational remodeling of RAC (Fig. 7c): Ssz1 is relocated to the PTE side of Zuo1 JD and could sample a variety of conformations, and most of these conformers are still incompatible with the binding of Ssb-ATP. The relocation of Ssz1 SBD close to the PTE facilitates the substrate-handover from Zuo1 to Ssz1, because the substrate binding site of Zuo1 ZHD is close to the N-terminus of Zuo1-ND that is interacting with Ssz1-SBD (Fig. 6d, e). Therefore, the nascent chain would replace the

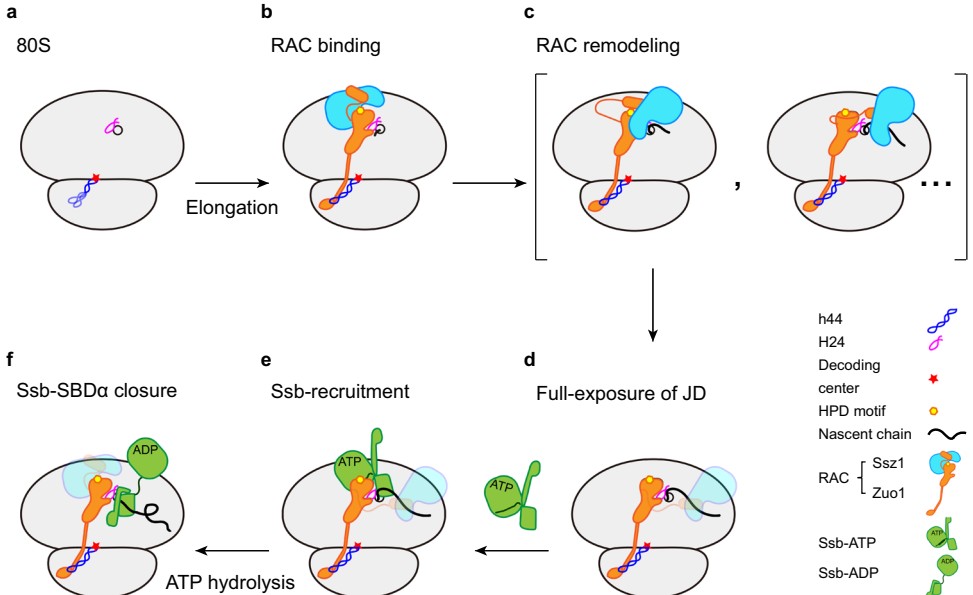

**Fig. 7 Model of RAC-Ssb actions in co-translational folding. a, b** RAC binds to the 80S ribosome, and through interactions of Zuo1 ZHD with H24-eL31-H101 (H24 as a key anchor point) on the 60S subunit and Zuo1 CTD with ES12-h44 on the 40S subunit. The JD of Zuo1 is initially covered by Ssz1 SBD. During translation elongation, the nascent chain emerges from the PTE (∼5 residues) and starts to interact with the hydrophobic patch on the N-terminus of ZHD-α3. **c** The elongated nascent chain triggers extensive conformational remodeling of RAC and Ssz1 is relocated to the PTE side of Zuo1 JD. A handover of substrate from Zuo1 to Ssz1 also occurs, which leads to further weakening of the Ssz1:Zuo1 interface. **d** The nascent chain competitively binds to Ssz1 SBD and thus disrupts the interface between Zuo1 and Ssz1 to mobilize Ssz1, resulting in a full exposure of the HPD motif. **e** Ssb-ATP recognizes Zuo1 JD and thus is recruited to the translating ribosome, with its SBD facing the PTE. **f** The nascent chain is handed over from RAC to Ssb. The ATP-hydrolysis of Ssb takes place, facilitated by Zuo1 JD, and Ssb transits to the ADP and high-affinity substrate-bound conformation.

conserved N-terminal LP motif in Zuo1 ND on Ssz1 SBD[34]. This leads to a disruption of the interface between the very N-terminal sequence of Zuo1 and Ssz1 and completely mobilizes Ssz1, which is still attached to Zuo1 through the first helix of Zuo1-ND (Fig. 4b), to fully expose Zuo JD (Fig. 7d). Subsequently, Ssb-ATP recognizes the exposed JD of Zuo1 and establishes a common HSP40:HSP70 interface ready to engage with the HPD motif (Fig. 7e). At this stage, Ssb-SBDβ in its ATP-bound conformation is placed in the proximity of the PTE, enabling a low-affinity binding to the elongated nascent chain. Stimulated by the HPD motif of Zuo1, the ATP-hydrolysis on Ssb triggers large conformational changes, including a repositioning of the SBD relatively to the NBD and a closure of Ssb SBDα for high-affinity binding to the nascent chain (Fig. 7f). Ssb-ADP after these conformational changes loses its contact with Zuo1 and establishes its direct interaction with the ribosome through its Ssb SBD[27–29]. With a further growing of nascent chain, substrate-engaged Ssb-ADP may completely detach from the ribosome, and this is likely the reason that we did not observe solid density for Ssb1 in the structures from the RNC-RAC-Ssb1 dataset. In cases of substrates with multiple Ssb binding motifs, Zuo1 could work in a cyclic manner to load multiple Ssb on a single peptide chain.

**Implication of RAC in other steps of translation regulation.** Unexpectedly, the endogenous 80S-RAC complexes purified through Zuo1 are greatly enriched in Stm1 and also contains eIF5A and eEF2. Stm1 is required for optimal translation under nutrient stress[49] and was initially proposed to be a ribosome preservation factor during quiescence and recovery[50]. Previously, it was found that 80S ribosomes from glucose-starved yeast cells contain both Stm1 and eIF5A[51]. A simultaneous association of both eEF2 and Stm1 with 80S ribosomes was also reported in the endogenous inactive or hibernating ribosomal complexes from other eukaryotic species[52–54]. However, this is not the reason for such an enrichment

of eEF2, eIF5A and Stm1 in our endogenous 80S-RAC complexes, because during our sample preparations of more than three biological replicates, the cells were cultured in nutrient-rich media and harvested in mid-log phase. Similar to our finding, very recently, Stm1 and eEF2 were also found to be in a structure of thermophilic 80S ribosomes purified through non-starvation protocol[55]. In addition, Stm1 was also detected in our RNC-RAC samples. The physiological relevance of this observation is not clear. One has to note that both Stm1 and eIF5A have multiple, yet-to-be fully characterized roles in different phases of translation. Stm1 also associates with polysomes and may function on actively translating ribosomes as well[56,57]. As to eIF5A, besides its role in facilitating the elongation of polyproline peptides[58], several functions outside elongation cycle, such as promoting efficient termination[59–61] and maintaining the fidelity of translation initiation[62], were also proposed. The fact that RAC is also implicated in regulating premature translation termination[16,18–20] seems to suggest that these factors could function in the same pathway.

One possible scenario for the formation of the Stm1/eIF5A-containing ribosome-RAC complex is (1) the cleavage of ester-bond on peptidyl-tRNA probably through premature termination; (2) the drop-off of both tRNA and mRNA from the ribosome; and (3) the binding of Stm1. Since Stm1 overlaps with both A-site tRNA and mRNA on the ribosome[63], it may also directly facilitate the release of tRNAs and mRNA. Our structures of RNC-RAC-Ssb1 dataset show that Zuo1 4HB deforms ES12 in a specific way that would pose a tension to h44. Zuo1 could potentially affect the conformational dynamics of the decoding center to kinetically modulate the decoding and termination events, and thus plays a role in preventing termination readthrough or promoting −1 frameshifting[16–21]. Therefore, RAC and eIF5A might work together to regulate translation termination or premature termination in certain conditions. And this under-characterized function of RAC may not be independent from its role in co-translational folding.

## Methods

**Yeast strains and plasmids**. A yeast strain (BY4741 background) expressing Zuo1 with a C-terminal 3X-FLAG tag was constructed by homologous recombination[64]. In brief, a nucleotide fragment containing the coding sequence of GS-linker-(3X-FLAG)-stop-codon, followed by HIS3MX6 gene and TEF terminator, was amplified from a modified pFA6a plasmid, and transformed into yeast cells to insert into the genomic position upstream the stop codon of ZUO1. The transformed cells were grown on the selective plate and the positive single colony was confirmed by sequencing. A deletion strain (BY4741 background) lacking DOM34 ($\Delta dom$) was similarly constructed.

Plasmids encoding Ssb1 substrate for purification of RNC were constructed based on the pYES2 backbone. GAL1 promoter in the pYES2 plasmid (high copy) was replaced by CUP1 promoter and an eGFP coding sequence was inserted into MCS (CUP1G plasmid). Fragments coding Pmt1(1-73) or Pmt1(1-52) were amplified from *S. cerevisiae* genomic DNA (S288C) using an upstream primer containing a strep-tag coding sequence, and inserted into the CUP1G plasmid upstream eGFP coding sequence, resulting in the Strep-Pmt1$_{1-73}$ (Pmt plasmid) and Strep-Pmt1$_{1-52}$ constructs, respectively. By plasmid PCR, Pmt1$_{71-73}$ in Pmt plasmid was substituted by Pro-Pro (CCG CCG) or Pro-Pro-Pro (CCT CCA CCT) codons (named as PP and PPP plasmids), and a CMV arrest peptide encoding sequence (GAA CCG CTG GTG CTG AGT GCG AAA AAA CTG AGC AGC CTG CTG ACC TGC ACA TAT ATT CCT CCT) was inserted downstream Pmt1 sequence in Strep-Pmt1$_{1-52}$ plasmid (named as CMV plasmid) (Supplementary Fig. 5a). Pmt, PP, PPP and CMV plasmids were transformed into wild type (BY4741) or DOM34 deletion ($\Delta dom$) strains for expression (wt-Pmt, wt-PP, wt-PPP, wt-CMV, $\Delta dom$-Pmt, $\Delta dom$-PP, $\Delta dom$-PPP and $\Delta dom$-CMV).

A deletion strain (BY4741 background) lacking ZUO1 ($\Delta zuo$) and two mutant strains with introduced point mutations (Supplementary Fig. 11) on genomic RPL31A/B genes were obtained by CRISPR/Cas9 and confirmed by sequencing. A genomic fragment of the ZUO1 gene (from 1000 bp upstream ZUO1 to 500 bp downstream ZUO1) was PCR-cloned from the *S. cerevisiae* genomic DNA (S288C) and inserted into pRS315 plasmid (single copy), thus creating vector version of ZUO1 under its endogenous promoter (pRS315-ZUO1). Site mutations and truncation of ZUO1 were constructed by plasmid PCR of pRS315-ZUO1. pRS315 plasmids carrying wild type ZUO1 or ZUO1 mutants were transformed into the $\Delta zuo$ strain. Cells were grown in SC-Leu medium to an $OD_{600}$ of ~0.8 and spotted as a ten-fold dilution series on YPD plates and grown at 18 °C for 72 h, or on paromomycin (200 μg/ml) containing YPD plates at 30 °C for 36 h.

**Purification of recombinant RAC and Ssb1**. Recombinant RAC was purified as described in our previous work[37]. Coding fragment of Ssb1 was amplified from *S. cerevisiae* genomic DNA (S288C), and cloned into the vector pET28a. The plasmid was transformed into *Escherichia coli* BL21 (DE3) cells for overexpression. Cells were induced at 18 °C with 0.4 mM IPTG for 12 h, harvested, resuspended in buffer A (20 mM HEPES-KOH pH 7.5, 500 mM KCl, 10 mM MgCl$_2$, 1 mM PMSF, and 20 mM imidazole) and subjected to ultrasonic lysis. The cell lysates were then clarified by centrifugation at 30,970 x g for 30 min at 4 °C in a JA 25.50 motor (Beckman Coulter). Supernatants were loaded onto a Ni-NTA column (GE Healthcare) and eluted with buffer B (20 mM HEPES-KOH pH 7.5, 120 mM KCl, 10 mM MgCl$_2$, and 250 mM imidazole). Eluates were further purified with a Resource Q column (1 ml, GE Healthcare). The Ssb1-containing fractions were pooled, concentrated and loaded onto a pre-equilibrated Superdex 200 column (10/300 GL, GE Healthcare) with buffer C (20 mM HEPES-KOH pH 7.5, 100 mM KCl, 10 mM MgCl$_2$).

**Endogenous 80S-RAC complex purification**. Cells were grown in 4 L YPD medium to an $OD_{600}$ of 2.0, harvested and washed with 50 ml ice-cold lysis buffer (40 mM HEPES-KOH pH 7.5, 100 mM KCl, 10 mM MgCl$_2$). Cell pellet was resuspended in lysis buffer of equal volume supplied with 0.05% NP40 and protease inhibitor cocktail (cOmplete ULTRA Tablets, Mini, EDTA-free, EASY-pack, Roche) and disrupted by vortexing with acid-washed glass beads in 50 ml tube. Cell lysate was clarified by centrifugation at 30,970 x g for 45 min at 4 °C in a JA 25.50 motor (Beckman Coulter) and the supernatant was incubated with pre-equilibrated Anti-FLAG M2 Affinity Gel (Sigma) for 90 min at 4 °C. The beads were then washed with 10 ml lysis buffer for three times and the samples were competitively eluted with 100 μg/ml flag peptides dissolved in lysis buffer for 30 min at 4 °C. The elution was concentrated with Amicon Ultra centrifugal filter units (Millipore) at 3000 g to an $A_{260}$ of ~12 for cryo-EM grid preparation. Samples were also examined by SDS-PAGE, and selected bands were subjected to mass spectrometry (Supplementary Fig. 1a).

**Purification of RNC complexes**. The N-terminal sequence of Pmt1, which harbors a strong Ssb-binding motif[14], was chosen as the model nascent chain. To synchronize nascent chains, several translation stalling motifs were introduced to the C-terminus of Pmt1 peptide, including a di-proline peptide encoded by two rare CCG codons (PP), a triple-proline peptide encoded by regular proline codons (PPP) and a well characterized cytomegalovirus arrest peptide (CMV) (Supplementary Fig. 5a). For ease of detection, sequences encoding Strep tag and eGFP were added to the 5′ and 3′-ends of these constructs. Since the PP and CMV are both

hard stalling signals, they might have elicited the ribosome-associated quality control (RQC) machinery[38,39] to eliminate the nascent chain and mRNA. Therefore, a mutant yeast strain lacking Dom34 ($\Delta dom$) was constructed. The two types of yeast cells transformed with various plasmids (wt-Pmt, wt-PP, wt-PPP, wt-CMV, $\Delta dom$-Pmt, $\Delta dom$-PP, $\Delta dom$-PPP and $\Delta dom$-CMV) were examined by live cell eGFP imaging and western blotting for the expression of engineered nascent chains (Supplementary Fig. 5b, c). Live cell eGFP imaging showed that the expression levels of these constructs in WT cells were all very low, even for the Pmt construct. The eGFP signals for the PP and CMV constructs were nearly undetectable. Western blotting using anti-strep antibody confirmed no strep signal was detected for the PP and CMV constructs. Importantly, western blotting also reported the presence of nascent chain-tRNA (NC-tRNA) moieties as well as the full-length proteins for the Pmt and PPP constructs. In sharp contrast, both the eGFP imaging and western blotting experiments showed that the levels of full-length proteins as well as accumulated NC-tRNAs dramatically increased in $\Delta dom$ cells for all constructs (Supplementary Fig. 5b, c).

Two strains, $\Delta dom$-PPP and $\Delta dom$-CMV, were used for the test of large-scale RNCs purification. To maximize the occupancy of RAC on the RNCs, additional incubation with excessive recombinant RAC was also performed: RAC proteins were added to the crude ribosomal pellets during RNC purification. Specifically, cells were grown in 3 L SC-Ura medium to an $OD_{600}$ of 1.5 and induced by adding CuSO$_4$ solution to a final concentration of 100 μM. After cultivating for additional 40 min, cells were quickly chilled in ice bath, harvested and washed with 30 ml buffer D (40 mM HEPES-KOH pH 7.5, 150 mM KCl, 5 mM MgCl$_2$, 0.05% NP40 and 100 μg/ml cycloheximide). Cell pellet was resuspended with 20 ml buffer D supplied with 2X protease inhibitor cocktail, disrupted by freeze milling (Jingxin). The frozen cell lysate powder was fast thawed, clarified by centrifugation at 30,970 × g for 45 min at 4 °C. Then every 3 ml of the supernatant was loaded onto 0.5 ml cushion buffer E (buffer D with 1 M sucrose and 2X protease inhibitor cocktail) and centrifuged in TLA-110 rotor (Beckman) at 234,675 × g for 65 min at 4 °C. The supernatant was discarded and the ribosome pellet was dissolved in 0.4 ml buffer F (40 mM HEPES-KOH pH 7.5, 100 mM KCl, 5 mM MgCl$_2$, 0.05% NP40, 100 μg/ml cycloheximide, 2X protease inhibitor cocktail and 6.4 μM RAC complex) per tube for 45 min at 4 °C with gentle shaking. The suspension was incubated with pre-equilibrated Strep-Tactin Sepharose (50% suspension, IBA) for 90 min at 4 °C. The beads were washed with 1.5 ml buffer G (buffer F without NP40, protease inhibitor cocktail and RAC complex) for five times and the bound RNCs were eluted with 0.5 ml buffer H (5 mM D-desthiobiotin dissolved in buffer G) for three times. The elution was concentrated with Amicon Ultra centrifugal filter units (Millipore) at 3,000 g to an $A_{260}$ of ~10. Purified samples were subjected to SDS-PAGE and mass spectrometry (Supplementary Fig. 1b). Western blotting analysis confirmed the presence of nascent chain-tRNAs in the RNC complexes (Supplementary Fig. 5c). As indicated by SDS-PAGE and MS analyses (Supplementary Fig. 1b), these RNC complexes were also enriched in factors of co-translational folding, including Zuo1, Ssz1 and Ssb1/2. Additional factors, such as eEF2 and other ribosome-binding proteins were also present in the sample. Considering that the PPP construct is more natural, the following structural analysis was done with RNC complexes from $\Delta dom$-PPP cells.

**Cryo-EM data collection**. Holey carbon grids (Quantifoil R1.2/1.3) were coated with a thin layer of freshly prepared carbon and glow-discharged with a plasma cleaner for 30 s (PDC-32G-2, Harrick Plasma). For the endogenous 80S-RAC sample, 4-μL aliquots of freshly purified sample (~0.2 μM) were placed onto grids, blotted for 1 s with a blotting force of −1, and rapidly plunged into liquid ethane using a FEI Vitrobot MarkIV operated at 6 °C and 100% humidity. The cryo-grids were screened in an FEI Talo Arctica and transferred to an FEI Titan Krios operated at 300 kV for data collection. The data were acquired using SerialEM[65] on a K2 Summit direct electron detector (Gatan) at a magnification of 130,000 (pixel size of 1.055 Å at the object scale), and with the defocus varying from −1.0 to −1.9 μm, using a total exposure of 8 s fractionated into 32 frames with a dose rate of ~8 electrons per pixel per second and a total exposure dose of ~58 e− per Å$^2$.

For the RNC-RAC sample, the purified RNC-RAC was supplemented with RAC (1:4) on ice for 45 min before cryo-grid preparation. The images were acquired using EPU software on a K3 Summit direct electron detector (Gatan) at a magnification of 81,000 (pixel size of 1.07 Å at the object scale), and with the defocus varying from −0.8 to −1.6 μm, using a total exposure of 3.84 s fractionated into 32 frames with a dose rate of ~14 electrons per pixel per second and a total exposure dose of ~47 e− per Å$^2$.

For the RNC-RAC-Ssb1 sample, the purified RNC-RAC was first supplemented with RAC (1:4) on ice for 10 min, and further incubated with Ssb1 (in the presence of 20-fold excess of ATP) on ice for 60 min. The final ratio of RNC:RAC:Ssb1:ATP was roughly 1:5:5:100. The images were acquired using SerialEM[65] on a K2 Summit direct electron detector (Gatan) at a magnification of 105,000 (pixel size of 1.356 Å at the object scale), and with the defocus varying from −1.0 to −1.6 μm, using a total exposure of 6.4 s fractionated into 32 frames with a dose rate of ~10 electrons per pixel per second and a total exposure dose of ~35 e− per Å$^2$.

**Data processing and analysis**. Movie stacks were summed and corrected (drift correction and dose weighting) using MotionCor2[66]. Contrast transfer function parameters (CTF) were estimated with Gctf program[67]. Image processing,

including micrograph manual screening, 2D and 3D classification, 3D auto-refine, CTF refinement and post-processing, were done with RELION 3.0 and 3.1[68].

For the endogenous 80S-RAC sample (Supplementary Fig. 2), 5,404 micrographs were used for particle auto-picking. After two rounds of 2D classification, 152,200 particles were kept and subjected to the initial round of 3D classification to discard bad particles, resulting in a dataset of 145,398 particles for further processing. One round of 3D refinement was done to provide a general alignment for all particles. Three parallel independent 3D classification procedures were applied, focusing on different parts of the 80S-RAC complex, including the PTE-RAC, L1 stalk/E-site, and eEF2 regions. First, the signal of RAC-PTE region was subtracted using a soft mask and the subtracted particles were subjected to one round of 3D classification (skip alignment, T = 10) with a soft mask of RAC. Among the resulting six classes, RAC displayed different conformations on the 60S subunit, and three of them (A1, A2 and A3) showed apparently more complete density for RAC. One class (23,667, 16.3% of total particles, State A2) showed the best-defined density of RAC. This class was re-extracted to obtain the whole 80S-RAC complex and was refined with a soft mask to an overall resolution of 3.4 Å (Map 1.2) according to the gold standard FSC 0.143 criterion. The density of Zuo1-NTD was further improved to 5–8 Å with sufficiently resolved secondary structural features by focused 3D refinement with a soft local mask (Map 1.4). Second, a soft mask of the L1 stalk/E-site region was used for focused classification (skip alignment, T = 10), yielding three classes with different L1 stalk positions, including the open, E-site tRNA-bound and eIF5A-bound states. Specifically, around 40% of particles are bound with eIF5A while only 11% of particles contain E-site tRNA. Third, a soft mask of eEF2 was used for the first round of focused classification (skip alignment, T = 10), resulting in the two meaningful classes, representing the eEF2-free and eEF2-bound states. The eEF2-free (77% of particles) was further split into four classes, with the ribosomes in differently rotated conformations. However, the 3D structure of the eEF2-containing particles does not contain P-site or A/P-site tRNA, and the ribosomal E-site is partially occupied. Therefore, a further 3D classification of this class on the L1/E-site region was applied, which revealed an 80S structure (9% of particles) in fully rotated conformation, with stable binding of eEF2, Stm1 and eIF5A (Fig. 1; Supplementary Fig. 2, Map 1.7).

For the RNC-RAC-Ssb1 sample (Supplementary Fig. 6), 5239 micrographs were used for particle auto-picking. After two rounds of 2D classification, 637,372 particles were kept and subjected to the initial round of 3D classification to discard bad particles, resulting in a dataset of 595,388 particles for further processing. One round of 3D refinement was done to provide a general alignment for all particles. The dataset was further split into eight classes. Two classes (44,892, 7.5% of total particles) displayed di-ribosome features, with the dimer structures centered at the positions of the stalled and colliding ribosomes, respectively. The orientation between the two 80S ribosomes is in general similar to the previously reported structures of di-ribosomes[69–72]. In particular, our dimer structure closely matches with the structure of the leading two ribosomes in the tri-ribosome stalled on the CGA-CGA codons on SDD1 mRNA[72], with the stalled and colliding ribosomes in non-rotated (A- and P-site tRNAs) and rotated (A/P- and P/E-site tRNAs) states, respectively. Nevertheless, most of the RNC-RAC particles (89%) are monosomes, spread in five 3D classes. Of these five classes (527,133 particles), four similar classes in intermediate state were subjected to two rounds of focused classification (skip alignment, T = 10), using a soft mask of the intersubunit space. Based on these classification results, the occupancies of tRNAs in monosomes are: no tRNA (94,346, 17.9% of monosomes), P-site tRNA only (44,454, 8.4% of monosomes), A/P + P/E-site tRNAs (71,374, 13.6% of monosomes) and A + P*-site tRNA (316,959 particles, 60.1% of monosomes). Very intriguing, the first class (94,346 particles) bears no tRNA but nascent chain, and further refinement of this class (Map 2.8, Stm1-engaged) also discovered the presence of Stm1 in the 80S ribosome (Supplementary Fig. 8c). Comparison with the translating RNC revealed a very dramatic conformation change for the head of the 40 S subunit in the Stm1-engaged ribosome, featuring a large swiveling movement up to 12° (Supplementary Fig. 8d). In addition, all particles of monosomes were subjected to two parallel independent 3D classification procedures. First, the density of RAC-PTE region was improved in a similar way as done with the endogenous 80S-RAC sample, using signal-subtraction and focused classification (skip alignment, T = 10). Among the resulting six classes, three of them (B1, B2, and B3) showed apparently more complete density for RAC and one class (106,203 particles, 20.1% of monosomes, State B2) showed the best-defined density of RAC. This class was re-extracted to obtain the complete RNC-RAC particles and was refined with a soft mask to an overall resolution of 2.9 Å (Map 2.2) according to the gold standard FSC 0.143 criterion. The density of RAC was further improved by focused 3D refinement with a soft local mask (Map 2.4). Second, the signal of 4HB-ES12 region was subtracted using a soft mask and the subtracted particles were subjected to one round of 3D classification (skip alignment, T = 10) with a soft mask of 4HB. Among the resulting eight classes, one class (60,651 particles, 11.5% of monosomes) showed the best density of 4HB (4HB-bound). Four classes that contained highly fragmented or very weak signal of 4HB, were subjected to another round of 3D classification (skip alignment, T = 10), yielding one class (50,758 particles, 9.6% of monosomes) for the 4HB-free state of ES12. The groups of 4HB-bound and 4HB-free RNCs were re-extracted as complete particles, and subjected to a global 3D refinement to produce the final maps for the 4HB-bound (Map 2.5) and 4HB-free (Map 2.7) states. Both maps were determined at an overall resolution

of 3.1 Å. The EM density of the 4HB-ES12 region in the 4HB-bound RNCs was further improved by focused 3D refinement with a soft local mask (Map 2.6).

For the RNC-RAC sample (Supplementary Fig. 9), 8,455 micrographs were used for particle auto-picking. After two rounds of 2D classification, 1,166,195 particles were kept and subjected to 3D classification to discard bad particles, resulting in a dataset of 823,278 particles for further processing. One round of 3D refinement was done to provide a general alignment for all particles, then the particles were subjected to focused classification (skip alignment, T = 10) using a soft mask of the intersubunit space. The particles were classified and then recombined into two groups, tRNA-bound (489,474 particles, 59.5%, elongating state) and Stm1-engaged (333,804 particles, 40.5%), for the following processing procedures. For the tRNA-bound particles, after one round of global 3D refinement, the signal of RAC-PTE region was subtracted and subjected to focused classification (skip alignment, T = 10) with a soft mask of RAC. Among the resulting six classes, three classes with more solid density of Zuo1 were subjected to another round focused classification (skip alignment, T = 10) using a soft mask of Ssz1. This cascade of focused classifications led to the identification of four general conformational groups for RAC on the ribosome: C1 (16,419 particles, 3.6% of tRNA-bound particles), C2 (11,605 particles, 2.4% of tRNA-bound particles), C3 (11,831 particles, 2.4% of tRNA-bound particles), and C4 (all remaining classes). In C4 maps, Zuo1-NTD is fairly resolved but no solid density of Ssz1 was found except those highly fragmented pieces near the PTE. Particles of the class C2 were re-extracted to obtain the complete RNC-RAC particles and were refined to an overall resolution of 3.3 Å (Map 3.4). For the Stm1-engaged particles, a similar processing strategy was used, resulting in one group of 18,195 particles (5.5% of Stm1-engaged particles) displaying a similar conformation to the class C2. These two classes from tRNA-bound and Stm1-engaged particles were pooled and refined to generate a density map at an overall resolution of 3.0 Å (Map 3.5). The EM density of RAC region of this map was further improved by focused 3D refinement with a soft local mask (Map 3.6). Upon sharpening, the interface between Ssz1-NBD and uL24 was resolved at side-chain level.

**Model building.** For the model building of the endogenous 80S-RAC and RNC-RAC complexes, the coordinates of the yeast 80S (PDB 4V8Y and 6TNU)[73,74] were used. The models of S. cerevisiae Zuo1, Ssz1 were predicted by AlphaFold2[45]. Model of S. cerevisiae Ssz1-ND$_{Zuo1}$ was built based on a crystallography study (PDB 6SR6)[34]. Models of individual components were docked as rigid bodies into the cryo-EM maps using USCF Chimera[75]. Model of ES12 (PDB 6TNU)[74] was fitted separately as a rigid body into the 4HB-bound and 4HB-free ES12 maps. Manual adjustments of Zuo1 and Ssz1 were done using COOT[76], in positions where the side chains of Zuo1 were resolved or partially resolved. Ssb1 (PDB 5TKY)[27] was docked to Zuo1 JD based on the coordinates of the JD$_{DnaJ}$-DnaK-ATP complex (PDB 5NRO)[46]. The putative substrate of Ssb1 was docked into Ssb1 SBD based on structural alignment to the structure of DnaK-SBDβ-peptide complex (PDB 4R5I)[48]. Structural analysis and figure preparation were done with Chimera or ChimeraX[75,77].

**Statistics and reproducibility.** Purification of the native 80S-RAC and RNC-RAC samples, the live cell eGFP imaging, the western blotting and the spot assay experiments were repeated for three times with similar results. No statistical analysis, other than those embedded in image processing, has been applied throughout the work.

**Reporting summary.** Further information on research design is available in the Nature Research Reporting Summary linked to this article.

## Data availability

The cryo-EM maps of the 80S-RAC (Map 1.2), RNC-RAC-Ssb1 (Map 2.2 and Map 2.5), and RNC-RAC (Map 3.4) dataset have been deposited in EMDB with accession codes EMD-32975, EMD-32977, EMD-32978, and EMD-32988, respectively. The cryo-EM map (Map 3.6) and atomic model of the RAC in the State C2 of the RNC-RAC dataset have been deposited in the EMDB and PDB with accession codes EMDB-32991 and PDB 7X3K, respectively. The raw data of the mass spectrometry analyses of the 80S-RAC and RNC-RAC samples are available as Supplementary Data 1. All materials developed in this study should be addressed to N.G.

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

## Acknowledgements
We thank the Core Facilities at the School of Life Sciences, Peking University for help with negative staining EM; the Electron Microscopy Laboratory and Cryo-EM Platform of Peking University for help with data collection; the High-performance Computing Platform of Peking University for help with computation. The work was supported by the National Science Foundation of China (31725007 to N.G. and 31922036 to N.L.), the National Key Research and Development Program of China (2019YFA0508904 to N.G.), and the Qidong-SLS Innovation Fund to N.G.

## Author contributions
Y.C. prepared the samples. Y.C. collected the cryo-EM data. Y.C., N.G. and N.L. performed EM analysis and model building. Y.C. and B.T. constructed the yeast mutants. N.G. and Y.C. wrote the manuscript.

## Competing interests
The authors declare no competing interests.
