## [Peer Review File · Nature Communications]

Structural remodeling of ribosome associated Hsp40-Hsp70 chaperones during co-translational foldingREVIEWER COMMENTS

Reviewer #1 (Remarks to the Author):

The group of Dr. Gao published the first cryo-EM structure of ribosome-associated complex (RAC) bound to ribosomes in 2014 (Zhang et al., 2014). This work was a breakthrough, which significantly promoted our understanding of RAC function and subsequent research in the field. Since, many cryo-EM/crystallography groups have tried to obtain higher resolution structures of ribosome-bound RAC/Ssb, however, as far as I know, nothing came out of these attempts.

This new manuscript by the Gao group provides long-awaited novel structural and functional details of ribosome-bound RAC. The publication of this work is of general importance and will strongly influence research in the field of translation and protein folding.

1. The authors present cryo-EM structures of RAC bound to non-translating as well as translating ribosomes.
2. The results provide novel insight into the ribosomal contact points of RAC and its role at the decoding center. The data identify functionally important contacts between h44-ES12 of the decoding center and the C-terminal end of the Zuo1 MD (an extended α -helix). Moreover, the structural data identify the contacts of Zuo1 to H101, H24, and Rpl31 at the amino acid /nucleotide level.
3. The authors present (preliminary) evidence how RAC cooperates with Ssb at the polypeptide tunnel exit. The findings reveal that RAC bound to non-translating ribosomes is in an autoinhibited state, in which the SDB of Ssz1 covers the Zuo1 J-domain. In contrast, RAC bound to translating ribosomes adopts a distinct conformation, in which the Zuo1 J-domain is free to interact with Ssb. Cryo-EM data combined with docking models provide initial (partially speculative) insight into how Ssb and RAC may cooperate.
4. The authors report the unexpected finding that Stm1 and eIF5 α are bound to RAC-ribosome complexes.

Major suggestions:

- I. The data are convincing and well presented. However, for the reasons described below I suggest to focus the main manuscript on 1 to 3.
- II. It is difficult to understand in some sections/Figures what is conclusions based on the data and what is models/speculation. I find the speculative parts intriguing and suggest to keep them in the manuscript. However, the manuscript should be reorganized, such that Results and Discussion are clearly separated.

Related to I.

The authors write: "... However, this is not the reason for such an enrichment of eIF5A and Stm1 in our endogenous 80S-RAC complexes, because during our sample preparations of more than three biological replicates, the cells were cultured in nutrient-rich media and harvested in mid-log phase".

In my opinion it is not sufficient to consider growth conditions. Important is also how cells are harvested. Here, harvesting for the purification of ribosomes via tagged Zuo1 was performed without addition of cycloheximide and was followed by centrifugation and cell washing. These conditions resemble glucose-depletion conditions, which lead to translational run-off (Ashe et al., 2000) and may well lead to the binding of e.g. Stm1 to ribosomes.

Suggested experiment: In case the authors wish to suggest that ribosome-RAC complexes isolated via tagged-Zuo1 are actively translating they should experimentally test this by performing ribosome profile analysis as described by Ashe and coworkers (Ashe et al., 2000). In these control experiments cell extracts should be prepared from the strain expressing tagged Zuo1, exactly as described for the purification of RAC-ribosome complexes. If the assumption is correct, ribosome profile analysis shall reveal a high concentration of polysomes and RAC as well as Stm1 and eIF5A should be in the polysome fractions.

Alternatively, one could transfer the paragraph "Stm1, eIF5A and eEF2 are present in the endogenous 80S-RAC complexes" into the Supplementary Material as a Supplementary Result and remove the last sentence of the abstract. Along the same line, I suggest to remove Stm1/eIF5A from the model in Figure 6. The eIF5A/Stm1 part distracts from the major findings. The findings indicate that RAC is indeed bound to empty ribosome-Stm1-eIF5A complexes, which is an interesting finding by itself. However, this seems to be a different story, which might be connected to the finding that Stm1 is also bound to RAC-RNC complexes, which lack a tRNA, however, still contain a nascent chain (Fig. 2).

Related to II.

The Results/Discussion connected to Fig. 5/Supplementary Figure 11 is partly speculative.

Fig. 5a- 5c data should remain in the Results part. In my view, Fig. 5a and 5b is solid data and allows for the conclusion that bound to non-translating 80S ribosomes Zuo1 cannot stimulate ATP-hydrolysis of Ssb, because Ssz1 conceals the J-domain. Fig. 5c shows an "uncharacterized density", which also should be part of the Results.

Fig. 5c-f and Supplementary Fig. 11 also include experimental data, however, also a lot of interesting speculation. These illustrations should be part of the Discussion.

Some suggestions/questions regarding Fig. 5c-f and Supplementary Fig. 11.

1. The authors argue that the "uncharacterized density" close to Zuo1 is not the nascent chain. I agree with this, as the nascent chain is too short to form this density. However, in my opinion this density could be a domain of Ssz1, which is no longer in the same position as in Fig. 5b. Probably, the Ssz1-SBD fits quite well into this density? Please explain, in case you wish to exclude this possibility.

2. Placement of Ssb into these structures is highly speculative. It is not even clear if the particles chosen for the structures in Fig. 5 and Fig. S11 contain Ssb. However, it is interesting to model possible contacts

of Ssb with the Zuo1 J-domain to get an idea if these complexes can be formed or are excluded due to steric clashes. Please mention these models in the Discussion.

3. Please comment on your view of the role of the Ssb-SBD² lid domain. The model in Fig. 5f suggests that the lid domain does not contribute to ribosome-binding of ATP-Ssb. However, two previous studies revealed that stable binding of Ssb to ribosomes critically depends on the SBD² (Gumiero et al., 2016; Hanebuth et al., 2016).

4. The authors suggest that the structures in Fig. 5 represent RNC-RAC-ADP-Ssb complexes. Could the authors speculate about the position of SBD² of ribosome-bound ADP-Ssb? By using the structural info of ADP-bound DnaK? Can SBD² be accommodated close to the ribosome when Ssb is in the ADP-bound conformation or does it clash with other components of the complex?

5. The authors state: "Our structural data are consistent with this notion: we did not observe densities for the main body of Ssb1, indicating that Ssb1-ADP is only flexibly attached to the ribosome through its SBD." Does this suggest that ADP-Ssb is only bound to the nascent chain? Or is there two contacts - one to the ribosome the other to the nascent chain?

Minor:

1. Concerning the model nascent chain: Pmt1 is a multiple membrane spanning protein localized in the ER membrane. The first transmembrane domain (TM) localized between residue 51 - 70. Such proteins are known to be targeted to the ER membrane co-translationally via the SRP-dependent pathway The CMV construct does not contain the TM, however, the PPP construct does (Figure S5). The TM inside the tunnel may recruit SRP. Was this observed? Why did the authors choose a membrane protein as a model nascent chain in this study? Please explain (e.g. in the Methods section).

2. Please mention in each Figure depicting RNCs, which nascent chain was used for purification.

3. Please unify the color code of the ribosomal subunits throughout the Figures.

4. In Fig. 2d: It is not clear where RACK1 (Asc1?) is shown in the structure (color code?) and why the label was added.

5. Spelling and grammar require some revision.

Ashe, M.P., De Long, S.K., and Sachs, A.B. (2000). Glucose depletion rapidly inhibits translation initiation in yeast. *Mol Biol Cell* 11, 833-848.

Gumiero, A., Conz, C., Gesé, G.V., Zhang, Y., Weyer, F.A., Lapouge, K., Kappes, J., von Plehwe, U., Schermann, G., Fitzke, E., et al. (2016). Interaction of the cotranslational Hsp70 Ssb with ribosomal proteins and rRNA depends on its lid domain. *Nat Commun* 7, 1-12.

Hanebuth, M.A., Kityk, R., Fries, S.J., Jain, A., Kriel, A., Albanese, V., Frickey, T., Peter, C., Mayer, M.P., Frydman, J., et al. (2016). Multivalent contacts of the Hsp70 Ssb contribute to its architecture on ribosomes and nascent chain interaction. *Nat Commun* 7, 13695.

Zhang, Y., Ma, C., Yuan, Y., Zhu, J., Li, N., Chen, C., Wu, S., Yu, L., Lei, J., and Gao, N. (2014). Structural basis for interaction of a cotranslational chaperone with the eukaryotic ribosome. *Nat Struct Mol Biol* 21, 1042-1046.

Reviewer #2 (Remarks to the Author):

The manuscript by Chen and co-workers represents a comprehensive structural analysis of Ssb-RAC on ribosomes from *Saccharomyces cerevisiae* employing mainly Cryo-EM techniques. RAC is a heterodimer composed of Zuo1 and Ssz. The analysed RAC-ribosome complexes (either 80S ribosomes or stalled RAC-containing ribosome-nascent chain complexes (RNCs)) were purified by affinity chromatography using either 3xFLAG (on the RAC component Zuo1) or Strep tags (on engineered stalled nascent chains), respectively.

In the first part, the structural data suggest an association of different factors, namely Stm1, eIF5A or eEF2 with Zuo1-containing 80S particles while only a minor occupation of the E- or P-site with tRNA in the ribosome was observed. The authors interpreted this as a novel potentially regulatory function of RAC beyond translation elongation and postulate a potential role of the above mentioned identified associated factors.

The investigation of RAC-RNCs in the second part revealed that the interactions between RAC and the ribosome seem to have rigid as well as flexible properties depending on the point of interaction. Furthermore, binding RAC-C-terminal domain can cause changes in the conformation of ribosomal helix h44, thereby potentially influencing translational dynamics in the decoding center. Based on their observations on the differences in binding of Ssz to RAC subunit Zuo1 before and after emergence of the polypeptide chain, a model is proposed in which Ssz dissociates from the Zuo1 J-domain while still being attached to the Zuo1 N-terminal domain upon binding of the nascent chain. As a consequence, the Ssz1-inhibited/blocked Zuo1 J-domain is released and can be bound by Ssb1 which ultimately positions Ssb close to the polypeptide exit tunnel for efficient binding to the nascent chain.

In summary, the manuscript contains a number of interesting findings, but many results are premature. Moreover, the results are presented in a reader unfriendly manner with too many technical details that are rather confusing. In the reviewer's opinion the manuscript needs extensive rewriting. For example, on multiple times details on the percentage/fraction of a certain particle class (after classification) are presented which is confusing and disturbs the readability of the text. It would help if the percentage of each particle class would instead be listed within in the figures with the summary of the particle classification and refinement (i.e. Supplementary Figures 2 and 6).

Importantly, the reviewer is not an expert in cryo-EM techniques and thus cannot judge on the quality of the experimental and technical data regarding cryo-EM, e.g. specific technical details regarding 3D classification, refinement strategies etc.

Major points:

- The manuscript consists of two main parts. The first one is predominantly based on mass spectrometry and cryo-EM analysis of purified endogenous tagged 80S-RAC particle (where Zuo1 is FLAG-tagged and 80S-RAC particles are affinity purified). The second part is mainly based on the purification and analysis of substrate-engaged 80S-RAC complexes with nascent chains. While the findings of part one, which include identification and visualization of an association of factors Stm1, eIF5A and eEF2 on these RAC complexes are novel and interesting, this part is too preliminary in the current state to warrant publication in a high-impact journal. In order to allow any conclusion about a physiological relevance of these "states", the findings clearly require to be substantiated by biochemical and genetic analyses. In the opinion of the reviewer this part of the manuscript should either be extensively shortened or these findings should be left out completely.

- The second part of the manuscript describes the structures of (engineered) 80S-RNC-RAC-(Ssb) complexes which provides the translating ribosome. The authors suggest a model of the molecular mechanism of how RAC is positioned at the ribosomal exit tunnel, how it might interact with the nascent chain and how the RAC complex ultimately positions Ssb correctly on the ribosome, e.g. with its SBD close to the polypeptide exit tunnel. The underlying different cryo-EM structures (structure models) are based on the analysis of substrate-engaged 80S-RAC complexes with a nascent chain. For these experiments RNC complexes were purified via an engineered Strep-tag at the N-terminus of the nascent chain and purified (endogenous) RNC are subsequently supplemented in vitro with recombinant RAC and Ssb-ATP. Delineated cryo-EM structures cover different states of the translating ribosome. Often the authors fit known crystal structures or structural models (e.g. of Ssb-ATP) into their cryo-EM structure and delineate a structure of the complex. Again, since the reviewer is no expert in cryo-EM it is difficult to judge how good the grounds are for such a fitting. However, what is exciting is that the derived structures provide first structural support for the previously proposed substrate hand-over model (of the nascent chain) from RAC to Ssb (Sinning & Rospert labs, Zhang, Gese et al., 2020 Nature Communications). The structures and the derived mechanistic model are plausible. However, since the

model (and the manuscript in general) is based almost exclusively on (partly fitted) structural data, the reviewer strongly feels that the new structural insights and ultimately the model of the substrate-hand over from RAC to Ssb itself require additional biochemical and/or genetic verification. In particular, certain postulated states/conformations of the remodeling of RAC on the ribosome upon engagement of the nascent chain as well as during hand-over of the nascent chain from RAC to Ssb necessitate biochemical or genetic verification. For example, postulated close proximity of certain residues of Zuo1 and Ssz1 or Zuo1 and Ssb could be probed by chemical cross-linking.

- The old nomenclature is used for ribosomal proteins. The new nomenclature for ribosomal proteins (Ban et al., 2014 *Curr. Opin. Struct. Biol.*) should be used throughout the manuscript.
- Some experimental details, e.g. the different types of stalling constructs and their description are too extensive in the main text and should be shortened. It would be easier to focus on the PPP stalling construct that was used for generation of the stalled RNC.
- What is the experimental proof for the blocking of the Zuo1 HPD motif by Ssz1 SBD? This has to be shown experimentally by mutations and crosslinking experiments.
- Can the postulated initial docking of Ssb on the RNC-RAC complex be verified experimentally (e.g. by chemical cross-linking)?
- What could be the function of the previously identified positively charged ribosome-binding region at the C-terminus of Ssb (in SBDalpha) in the model since in the “proposed” structure the Ssb C-terminus does not seem to contact the ribosome?
- The reviewer could not locate Supplementary Table 1
- Supplementary Figure 1A: According to the silver stained gel, it seems that a detectable amount of Ssb is also co-purified upon purification of tagged Zuo1 and ribosomes (also present in the pellet fraction indicating ribosome association). Did the authors detect Ssb in their 3D classification of the 80S-RAC particles?

Minor points:

- line 56: typo “addition” should be “additional”
- line 290: “Supplementary Fig. 8c” should be “Supplementary Fig. 8b”

Reviewer #3 (Remarks to the Author):

Structural insights into the function of ribosome associated Hsp40-Hsp70 complex in translation regulation by Chen et al. This manuscript investigates the molecular chaperone complex RAC and its cooperation with the Hsp70 Ssb during cotranslational protein folding. A particular goal is to describe the coordinated actions of RAC (Zuo1 and Ssz1) and Ssb1. To that end, the authors obtained cryo-EM structures of RAC bound to a nascent chain free ribosome and compared that to the structure of ribosome-nascent chain complexes. Overall they have a nice collection of different structures that contain different components of the translational machinery, and the structures provide more detailed information about how Zuo1 interacts with the ribosome. However, it isn't clear how the structures might progress from one complex to another and many of the steps in the model of Figure 6 are speculative, particularly when trying to discuss Ssb interaction and the presence of Stm1 and eIF5A and eEF2 in their complexes. Overall, the main impact of this manuscript is a more detailed interaction about Zuo1, and it does not directly answer larger questions of handoff of newly translated protein from RAC to Ssb.

Specific comments

In general, the writing is not concise, and in many cases assumes that the reader is very familiar with previous structural and functional analysis of RAC. For example, "This suggests that Zuo1 present a physical and functional link between the PTE and the decoding center through its ES12-h44 interacting CTD (4HB)" (line 91).

It is difficult to interpret whether the RAC-80S structures are authentic functional steps considering that the 80S ribosomes are likely in a resting state (lines 138-142 and lines 166-169.)

The authors describe 3 different conformational states of RAC on the 60S that show conformational changes in Zuo1 and it is difficult to assess which is 'correct'. One figure they focused on in Fig 1 Map 3.4 was based on only 9% of the particles. The RAC figures shown in Fig. 1, 2 and Supp 2 and 3 are somewhat poorly resolved and there is no obvious comparison to previously existing structures.

Supplementary Table 1 needs more description, such as details on which protein hits were significant and which ones were not.

There is also minimal validation of newly described structural interactions and that is limited to mutational analysis of Zuo1 and effect on growth. They make mutations in Zuo1 residues critical for ribosome interaction and show that they disrupt function in vivo but do not show that the mutations also disrupt ribosome interaction.

The native substrate used for the RNC-RAC structure has a lot of modifications, raising questions about whether it is really still a native substrate after adding translational stalling motifs and eGFP?

Fig 2 is a composite map of a representative RNC-RAC-Ssb1 complex built from the the 40S subunit from Map 5.1 and the 60S subunit plus tRNAs from Map 4.2, not an intact structure, and Ssb was further modeled onto that structure. Again, given the heterogeneity of the samples, it is difficult to assess whether these are authentic structures.

Reviewer #4 (Remarks to the Author):

This work focuses on the structural basis of 80S ribosome-binding by the Ribosome Associated co-chaperone complex (RAC). The authors purified and reconstructed multiple instances of budding yeast RAC-ribosome complexes, some of which reached sufficiently fine resolutions to model unknown aspects of the chaperone's structures and ribosome recognition motifs. In addition, by exploiting different tagging and purification strategies, the authors made surprising observations about 80S-RAC complexes with, and without, nascent polypeptide chains, tRNAs, and other translation-associated factors.

First, by tagging Zuo1, one of the main components of RAC, the authors semi-purified endogenous 80S-RAC complexes. Surprisingly, the ribosomes reconstructed from this sample lacked mRNA, tRNA, or nascent chains. Rather, the predominant species was an entirely unexpected 80S-RAC-eEF2-eIF5A-Stm1 particle. The authors speculate about the meaning of this particle--invoking potential roles for RAC in termination or hibernation--but without clear functional insights into the roles of the co-chaperone on ribosomes that lack nascent polypeptides.

Second, the authors then designed over-expression constructs with elongation-stalling properties and expressed these in Dom34 knockout yeast cells. Under these conditions, stalled ribosomes with nascent chains accumulate to high levels with bound RAC. The authors purified and reconstructed these complexes (80S-RNC-RCA) to identify different locations and conformations of Zuo1's domains relative to the exit site of the protein translocation tunnel. This set of structures suggest how Zuo1 flexibly engages the ribosome using multiple, adjacent, and semi-stable poses. By integrating prior genetic and structural studies, the authors propose how Zuo1 may recruit and regulate Ssb's substrate engagement and ATP hydrolysis cycle.

Together, this paper includes multiple observations of interest, including the possibility that RAC has a role with inactive Stm1-blocked ribosomes. Before publication, we recommend addressing the following issues:

Major points:

1. We would like to suggest the authors consider separating the experimental results and more speculative aspects of the discussion. Many statements in this manuscript are deduced from the references or are speculation. In this case, it seems important to guide the reader through the observations and save the more daring speculations for later. For example, is it possible the unexpected nature of the particle obtained by tagging and pulling down Zuo1 is due to the affinity tag on Zuo1? This seems as likely as the possibility the authors propose, that "...our endogenous 80S-RAC structure should resemble an initial state of RAC binding on the 80S prior to nascent chain engagement." Maybe, but with no tRNA, no mRNA, and no nascent chain?
2. There does not appear to be interpretable density for Ssb in the maps of the RNC-RCA-Ssb complexes- is Ssb present in the complex? Do the authors have a reconstruction of the RNC-RCA complex (without adding the exogenous Ssb)? It could be informative to compare the structures of RNC-RCA before and after adding Ssb.
3. Was Stm1 identified in the mass spectrometry? I did not see it annotated in the supplementary material. It looks like Stm1, but was the identification strictly from the cryoEM density?
4. Line 386-387. The authors claimed, "Ssz1 should remain attached to Zuo1 through the flexible N-terminal sequence of Zuo1 NTD.". Are there any experimental data or references to support this statement?
5. Atomic model building and validation. The models were primarily built from existing structures with rigid-body docking, although manual tuning in Coot is mentioned in the methods. Please provide standard validation metrics, for example from MOLPROBITY, or Phenix and structural statistics table.

Minor points:

1. Typographic errors on line 281 "displays both translational and rotational of movement relative to the 60S" and on line 339. "... several single lysine (K341A, K342, 344A and K348) or ..." should be "... several single lysine (K341A, K342A, K344A and K348A) or ..."
2. Fig. 5b. Please label the HPD motif in the figure.
3. Supplementary Figure 11c. Is it Ssb1-ADP or Ssb1-apo?
4. Please make sure that the EM density maps and models are uploaded to EMDB and PDB. Also please provide the PDB/EMDB numbers in the manuscripts.
5. The "substrate relay" model is mentioned in the abstract and cited in the discussion, but never explained. What is this model and how does it pertain to this study?

6. The manuscript uses the term “in vivo” in many instances that are engineered, like exogenous over-expression of a stalling nascent chain in a dom34 null background. It seems in vivo does not convey a valid meaning for these conditions.

Major updates in the revision

(1) We have collected another dataset on the complex of RNC-RAC (without supplementing Ssb1 in the sample). From this dataset, a collection of new structures with solid density for both Zuo1 and Ssz1 were obtained. One state was resolved for nearly full-length of Zuo1 and Ssz1.

(2) Both figures and text have been extensively revised, with Results and Discussion being clearly separated.

REVIEWER COMMENTS

Reviewer #1 (Remarks to the Author):

The group of Dr. Gao published the first cryo-EM structure of ribosome-associated complex (RAC) bound to ribosomes in 2014 (Zhang et al., 2014). This work was a breakthrough, which significantly promoted our understanding of RAC function and subsequent research in the field. Since, many cryo-EM/crystallography groups have tried to obtain higher resolution structures of ribosome-bound RAC/Ssb, however, as far as I know, nothing came out of these attempts.

This new manuscript by the Gao group provides long-awaited novel structural and functional details of ribosome-bound RAC. The publication of this work is of general importance and will strongly influence research in the field of translation and protein folding.

1. The authors present cryo-EM structures of RAC bound to non-translating as well as translating ribosomes.

2. The results provide novel insight into the ribosomal contact points of RAC and its role at the decoding center. The data identify functionally important contacts between h44-ES12 of the decoding center and the C-terminal end of the Zuo1 MD (an extended α -helix). Moreover, the structural data identify the contacts of Zuo1 to H101, H24, and Rpl31 at the amino acid /nucleotide level.

3. The authors present (preliminary) evidence how RAC cooperates with Ssb at the polypeptide tunnel exit. The findings reveal that RAC bound to non-translating ribosomes is in an autoinhibited state, in which the SDB of Ssz1 covers the Zuo1 J-domain. In contrast, RAC bound to translating ribosomes adopts a distinct conformation, in which the Zuo1 J-domain is free to interact with Ssb. Cryo-EM data combined with docking models provide initial (partially speculative) insight into how Ssb and RAC may cooperate.

4. The authors report the unexpected finding that Stm1 and eIF5 α are bound to RAC-ribosome complexes.

We sincerely thank this reviewer for the summary of our new findings and his/her encouraging comments.

Major suggestions:

I. The data are convincing and well presented. However, for the reasons described

below I suggest to focus the main manuscript on 1 to 3.

II. It is difficult to understand in some sections/Figures what is conclusions based on the data and what is models/speculation. I find the speculative parts intriguing and suggest to keep them in the manuscript. However, the manuscript should be reorganized, such that Results and Discussion are clearly separated.

In the revision, the manuscript has been reorganized to focus on the relatively known role of RAC in co-translational folding. We have also separated the sections of Results and Discussions. The finding of the co-purification of RAC, eIF5A and Stm1 and relative discussions were moved to the Discussion section.

Related to I.

The authors write: "... However, this is not the reason for such an enrichment of eIF5A and Stm1 in our endogenous 80S-RAC complexes, because during our sample preparations of more than three biological replicates, the cells were cultured in nutrient-rich media and harvested in mid-log phase".

In my opinion it is not sufficient to consider growth conditions. Important is also how cells are harvested. Here, harvesting for the purification of ribosomes via tagged Zuo1 was performed without addition of cycloheximide and was followed by centrifugation and cell washing. These conditions resemble glucose-depletion conditions, which lead to translational run-off (Ashe et al., 2000) and may well lead to the binding of e.g. Stm1 to ribosomes.

Suggested experiment: In case the authors wish to suggest that ribosome-RAC complexes isolated via tagged-Zuo1 are actively translating they should experimentally test this by performing ribosome profile analysis as described by Ashe and coworkers (Ashe et al., 2000). In these control experiments cell extracts should be prepared from the strain expressing tagged Zuo1, exactly as described for the purification of RAC-ribosome complexes. If the assumption is correct, ribosome profile analysis shall reveal a high concentration of polysomes and RAC as well as Stm1 and eIF5 α should be in the polysome fractions.

We thank the reviewer for this helpful suggestion. Ashe et al. (Mol Biol Cell, 2000) found that the removal of glucose from yeast medium led to a rapid inhibition of initiation (sharp 80S peak and dramatically reduced polysomes), and that this inhibition was independent of new transcription. In their experiments, yeast cells were first grown in YPD and resuspended in YP medium lacking glucose for different time ranges (1-10 min, and 10 min for most experiments), and then the cells were treated with CHX, lysed and analyzed by sucrose density gradient centrifugation. The wash, lysis and centrifugation buffers used in this study were all glucose-free.

In our experiments, we directly lysed the cells out of YPD medium and did not have the additional incubation in YPD or other types of medium. CHX is well known to lock the ribosomes in a certain conformation. This is the reason that we did not include CHX in our buffers in the first place. We agree with the reviewer that the following cell wash

and disruption was in a glucose-free condition. To test the effect of glucose and CHX in polysome preservation, we did the following experiment. We constructed a yeast strain by CRISPR/Cas9 with Stm1, Zuo1 and eIF5A labelled with different tags (BY4741 background, Stm1-C-Strep, eIF5A-N-HA, and Zuo1-C-FLAG), and performed ribosome profile analysis. The experiments were done in four conditions, with or without CHX, and with or without glucose in the wash/lysis/centrifugation buffers. The centrifugation fractions were examined by Western blotting with commercial antibodies against these tagged proteins. The results (Rebuttal Figure 1) are: (1) We did not observe significant impact of glucose on the distribution of ribosomal fractions; (2) As expected, CHX inhibited the “run-off” and significantly preserved polysome fractions; (3) All three factors are present in both the 80S and polysome fractions in these four conditions. The signal for eIF5A is relatively weak, compared with the other factors. But it is clear that eIF5A is present in the polysome fractions with a longer exposure. Notably, a very recent study from Sinning group also reported that Stm1 is present in the tRNA and eEF2-containing ribosomes (the ribosomes were purified without starvation protocol) (Kisonaite et al., 2022).

Rebuttal Figure 1 Ribosome profile analysis of yeast cells (BY4741 background, *Stm1*-C-Strep, *eIF5A*-N-HA, and *Zuo1*-C-FLAG). Exposure time, 0.1s.

Alternatively, one could transfer the paragraph "*Stm1*, *eIF5A* and *eEF2* are present in the endogenous 80S-RAC complexes" into the Supplementary Material as a Supplementary Result and remove the last sentence of the abstract. Along the same line, I suggest to remove *Stm1/eIF5A* from the model in Figure 6. The *eIF5A/eIF5A* part distracts from the major findings. The findings indicate that RAC is indeed bound to empty ribosome-*Stm1-eIF5A* complexes, which is an interesting finding by itself. However, this seems to be a different story, which might be connected to the finding that *Stm1* is also bound to RAC-RNC complexes, which lack a tRNA, however, still contain a nascent chain (Fig. 2).

The validation experiments confirmed that Zuo1, Stm1 and eIF5A all associate with the polysomes (Rebuttal Figure 1). We agree with the reviewer that this finding is beyond the scope of this study. Therefore, we have extensively revised the manuscript and the implications of this finding is only briefly described in the Discussion. Also, we have shortened the first section of the Results and changed the title to "Compositional and structural characterization of the endogenous 80S-RAC complexes affinity-purified through tagged Zuo1". The abstract was also largely revised.

Related to II.

The Results/Discussion connected to Fig. 5/Supplementary Figure 11 is partly speculative.

Fig. 5a- 5c data should remain in the Results part. In my view, Fig. 5a and 5b is solid data and allows for the conclusion that bound to non-translating 80S ribosomes Zuo1 cannot stimulate ATP-hydrolysis of Ssb, because Ssz1 conceals the J-domain. Fig. 5c shows an "uncharacterized density", which also should be part of the Results.

Fig. 5c-f and Supplementary Fig. 11 also include experimental data, however, also a lot of interesting speculation. These illustrations should be part of the Discussion.

In the revision, we have added a new cryo-EM dataset and reorganized the figures. The experimental results and speculations are now clearly separated in figures. The new model figure (Fig. 7) was also revised to focus on co-translational folding.

Some suggestions/questions regarding Fig. 5c-f and Supplementary Fig. 11.

1. The authors argue that the "uncharacterized density" close to Zuo1 is not the nascent chain. I agree with this, as the nascent chain is too short to form this density. However, in my opinion this density could be a domain of Ssz1, which is no longer in the same position as in Fig. 5b. Probably, the Ssz1-SBD β fits quite well into this density? Please explain, in case you wish to exclude this possibility.

The reviewer is correct that this density could also accommodate the SBD β of Ssz1 in terms of size. We putatively assigned it as the SBD β of Ssb because it is close to the PTE and next to the position of the SBD β of aligned Ssb-ATP structure.

In the revision, we have removed this discussion since we could not exclude other possibilities.

2. Placement of Ssb into these structures is highly speculative. It is not even clear if the particles chosen for the structures in Fig. 5 and Fig. S11 contain Ssb. However, it is interesting to model possible contacts of Ssb with the Zuo1 J-domain to get an idea if these complexes can be formed or are excluded due to steric clashes. Please mention these models in the Discussion.

The suggestion is well taken. We have now included them in the Discussion. The structural alignment was present in a revised Fig. 6.

3. Please comment on your view of the role of the Ssb-SBD α lid domain. The model in Fig. 5f suggests that the lid domain does not contribute to ribosome-binding of ATP-Ssb. However, two previous studies revealed that stable binding of Ssb to ribosomes critically depends on the SBD α (Gumiero et al., 2016; Hanebuth et al., 2016).

There is no conflict between our model and these two studies. Our Fig. 6a-b in the revision was prepared with aligned Ssb structure in the ATP-bound form. This only reflects the initial docking of Ssb-ATP to the ribosome. Once ATP hydrolysis occurs, the SBD α would fold back to clamp the substrate with the SBD β . At the point, Ssb should undergo large conformational changes, and its orientations to Zuo1-HPD and to the ribosome will both alter. These two previous studies (Gumiero et al., 2016; Hanebuth et al., 2016) used cell-based assays to detect the binding of Ssb and its mutants to the ribosome. Given that most of Ssb-ATP would be converted to Ssb-ADP in a cellular context, these findings likely imply that SBD α is critically required for the stable binding of Ssb-ADP to the ribosome. Indeed, very recently, Craig lab demonstrated that Ssb displays differential crosslinks with Ssz1 and the ribosome when in the ATP or ADP bound states. Ssb-ATP could be crosslinked with Ssz1-NBD, whereas Ssb-ADP directly interacts with the ribosome with a crosslink between SBD β and uL24. (Lee et al., 2021). Therefore, our model highly agrees with these published data.

4. The authors suggest that the structures in Fig. 5 represent RNC-RAC-ADP-Ssb complexes. Could the authors speculate about the position of SBD α of ribosome-bound ADP-Ssb? By using the structural info of ADP-bound DnaK? Can SBD α be accommodated close to the ribosome when Ssb is in the ADP-bound conformation or does it clash with other components of the complex?

We could not identify densities of Ssb in the maps from the RNC-RAC-Ssb1 dataset, except a small piece shown in the previous Fig. 5c (now removed in the revision). Based on the structures of ATP- and ADP-bound DnaK (PDB 5NRO, 2KHO), its interface for DnaJ would be greatly changed in the ADP-bound form. Therefore, Ssb-ADP should not bind or has very low affinity for Zuo1 JD. At this point, we do not have a reference to fit the structure of Ssb-ADP in our models. If we superposed the model of Ssb-ADP to our structure using the two domains of Ssb as references one would see steric clashes in both cases (Rebuttal Figure 2). First, if Ssb-NBD is used as the reference of alignment (assuming after ATP hydrolysis, the relative orientation between Ssb-NBD and Zuo1-JD does not change), the SBD of Ssb-ADP would partially overlap with Zuo1 JD. Second, if Ssb-SBD β is used as the reference of alignment, the SBD α of Ssb-ADP would clash with the ribosome.

Rebuttal Figure 2. Structural alignment of Ssb1-ADP with Ssb1-ATP, using Ssb NDB (a) or SBD β (b) as reference. The docking of Ssb1-ATP on Zuo1 JD is based on the related studies of HSP40:HSP70 interaction.

Therefore, Ssb-ADP should bind to the ribosome in a different orientation from the initial position of Ssb-ATP. According to the published data (Gumiero et al., 2016; Hanebuth et al., 2016; Lee et al., 2021), the position of Ssb-SBD in the ADP-bound conformation should be roughly close to the PTE and near uL24, uL29, uL23, eL19 and ES24.

5. The authors state: "Our structural data are consistent with this notion: we did not observe densities for the main body of Ssb1, indicating that Ssb1-ADP is only flexibly attached to the ribosome through its SBD." Does this suggest that ADP-Ssb is only bound to the nascent chain? Or is there two contacts - one to the ribosome the other to the nascent chain?

We have revised this statement to avoid confusion. In the field, a prevailing model is that multiple Ssb molecules could bind to the same nascent chain. Therefore, it is likely upon the growing of nascent chains, newly bound Ssb-ADP loses its direct contact with the ribosome.

Minor:

1. Concerning the model nascent chain: Pmt1 is a multiple membrane spanning protein localized in the ER membrane. The first transmembrane domain (TM) localized between residue 51 - 70. Such proteins are known to be targeted to the ER membrane co-translationally via the SRP-dependent pathway. The CMV construct does not contain the TM, however, the PPP construct does (Figure S5). The TM inside the tunnel may recruit SRP. Was this observed? Why did the authors choose a membrane protein as a model nascent chain in this study? Please explain (e.g. in the Methods section).

We did not observe the SRP components in the structures and in the data of mass spectrometry. We chose the N-terminal sequence of Pmt1 as the nascent chain

because a previous study of Ssb-nascent chain interactions revealed that Pmt1 had a very sharp Ssb-binding peak in its N-terminal sequence (Rebuttal Figure 3) (Doring et al., 2017). Taking this sequence as the nascent chain might give us the maximal efficiency to obtain the suitable RNC (substrate) for RAC-Ssb. Notably, this sequence is also among the most enriched recognition sites of Ssb (about 14-fold). We have realized that it is a membrane protein. Since it was detected to be a highly enriched substrate of Ssb, we reasoned that it might still be a good candidate for our purpose. In addition, Pmt1 does not have a dedicated signal peptide before its first TM (51-70). Therefore, the first TM might be its SRP recognition site (Strahl-Bolsinger and Scheinost, 1999) and our construct would not evoke the SRP system since TM1 is still in the tunnel.

Rebuttal Figure 3. Ssb binding sites within the N-terminal sequence of Pmt1 (adapted from Doring et al., 2017). The X-axis is P-site codon position from ribosome footprint sequencing.

2. Please mention in each Figure depicting RNCs, which nascent chain was used for purification.

Only the construct of Δdom -PPP was used for cryo-EM analysis of the RNC-RAC complex. We have indicated this point clearly in the Methods in the revision.

3. Please unify the color code of the ribosomal subunits throughout the Figures.

Suggestion is taken. We have changed the colors of certain figures. In other figures, we used different colors to highlight that these structures are from different density maps.

4. In Fig. 2d: It is not clear where RACK1 (Asc1?) is shown in the structure (color code?) and why the label was added.

We labeled Asc1 to help readers with the orientation. The figure has been revised.

5. Spelling and grammar require some revision.

Thanks. We have double checked the grammar and spelling.

Ashe, M.P., De Long, S.K., and Sachs, A.B. (2000). Glucose depletion rapidly inhibits translation initiation in yeast. *Mol Biol Cell* 11, 833-848.

Gumiero, A., Conz, C., Gesé, G.V., Zhang, Y., Weyer, F.A., Lapouge, K., Kappes, J., von Plehwe, U., Schermann, G., Fitzke, E., et al. (2016). Interaction of the cotranslational Hsp70 Ssb with ribosomal proteins and rRNA depends on its lid domain. *Nat Commun* 7, 1-12.

Hanebuth, M.A., Kityk, R., Fries, S.J., Jain, A., Kriel, A., Albanese, V., Frickey, T., Peter, C., Mayer, M.P., Frydman, J., et al. (2016). Multivalent contacts of the Hsp70 Ssb contribute to its architecture on ribosomes and nascent chain interaction. *Nat Commun* 7, 13695.

Zhang, Y., Ma, C., Yuan, Y., Zhu, J., Li, N., Chen, C., Wu, S., Yu, L., Lei, J., and Gao, N. (2014). Structural basis for interaction of a cotranslational chaperone with the eukaryotic ribosome. *Nat Struct Mol Biol* 21, 1042-1046.

Reviewer #2 (Remarks to the Author):

The manuscript by Chen and co-workers represents a comprehensive structural analysis of Ssb-RAC on ribosomes from *Saccharomyces cerevisiae* employing mainly Cryo-EM techniques. RAC is a heterodimer composed of Zuo1 and Ssz. The analysed RAC-ribosome complexes (either 80S ribosomes or stalled RAC-containing ribosome-nascent chain complexes (RNCs) were purified by affinity chromatography using either 3xFLAG (on the RAC component Zuo1) or Strep tags (on engineered stalled nascent chains), respectively.

In the first part, the structural data suggest an association of different factors, namely Stm1, eIF5A or eEF2 with Zuo1-containing 80S particles while only a minor occupation of the E- or P-site with tRNA in the ribosome was observed. The authors interpreted this as a novel potentially regulatory function of RAC beyond translation elongation and postulate a potential role of the above mentioned identified associated factors.

The investigation of RAC-RNCs in the second part revealed that the interactions between RAC and the ribosome seem to have rigid as well as flexible properties depending on the point of interaction. Furthermore, binding RAC-C-terminal domain can cause changes in the conformation of ribosomal helix h44, thereby potentially influencing translational dynamics in the decoding center. Based on their observations on the differences in binding of Ssz to RAC subunit Zuo1 before and after emergence of the polypeptide chain, a model is proposed in which Ssz dissociates from the Zuo1 J-domain while still being attached to the Zuo1 N-terminal domain upon binding of the nascent chain. As a consequence, the Ssz1-inhibited/blocked Zuo1 J-domain is

released and can be bound by Ssb1 which ultimately positions Ssb close to the polypeptide exit tunnel for efficient binding to the nascent chain.

In summary, the manuscript contains a number of interesting findings, but many results are premature. Moreover, the results are presented in a reader unfriendly manner with too many technical details that are rather confusing. In the reviewer's opinion the manuscript needs extensive rewriting. For example, on multiple times details on the percentage/fraction of a certain particle class (after classification) are presented which is confusing and disturbs the readability of the text. It would help if the percentage of each particle class would instead be listed within in the figures with the summary of the particle classification and refinement (i.e. Supplementary Figures 2 and 6).

Suggestion is well taken. We have extensively revised both the figures and text. Also, we have reduced the technical explanation on imaging processing in the main text.

Importantly, the reviewer is not an expert in cryo-EM techniques and thus cannot judge on the quality of the experimental and technical data regarding cryo-EM, e.g. specific technical details regarding 3D classification, refinement strategies etc.

Major points:

- The manuscript consists of two main parts. The first one is predominantly based on mass spectrometry and cryo-EM analysis of purified endogenous tagged 80S-RAC particle (where Zuo1 is FLAG-tagged and 80S-RAC particles are affinity purified). The second part is mainly based on the purification and analysis of substrate-engaged 80S-RAC complexes with nascent chains. While the findings of part one, which include identification and visualization of an association of factors Stm1, eIF5A and eEF2 on these RAC complexes are novel and interesting, this part is too preliminary in the current state to warrant publication in a high-impact journal. In order to allow any conclusion about a physiological relevance of these "states", the findings clearly require to be substantiated by biochemical and genetic analyses. In the opinion of the reviewer this part of the manuscript should either be extensively shortened or these findings should be left out completely.

We thank the reviewer for this suggestion. We have extensively shortened these discussions and clearly separated the experimental results and speculation/hypothesis in the revision. Please also refer to our response to related questions from Reviewer 1.

- The second part of the manuscript describes the structures of (engineered) 80S-RNC-RAC-(Ssb) complexes which provides the translating ribosome. The authors suggest a model of the molecular mechanism of how RAC is positioned at the ribosomal exit tunnel, how it might interact with the nascent chain and how the RAC complex ultimately positions Ssb correctly on the ribosome, e.g. with its SBD close to

the polypeptide exit tunnel. The underlying different cryo-EM structures (structure models) are based on the analysis of substrate-engaged 80S-RAC complexes with a nascent chain. For these experiments RNC complexes were purified via an engineered Strep-tag at the N-terminus of the nascent chain and purified (endogenous) RNC are subsequently supplemented in vitro with recombinant RAC and Ssb-ATP. Delineated cryo-EM structures cover different states of the translating ribosome. Often the authors fit known crystal structures or structural models (e.g. of Ssb-ATP) into their cryo-EM structure and delineate a structure of the complex. Again, since the reviewer is no expert in cryo-EM it is difficult to judge how good the grounds are for such a fitting. However, what is exciting is that the derived structures provide first structural support for the previously proposed substrate hand-over model (of the nascent chain) from RAC to Ssb (Sinning & Rospert labs, Zhang, Gese et al., 2020 Nature Communications). The structures and the derived mechanistic model are plausible. However, since the model (and the manuscript in general) is based almost exclusively on (partly fitted) structural data, the reviewer strongly feels that the new structural insights and ultimately the model of the substrate-hand over from RAC to Ssb itself require additional biochemical and/or genetic verification. In particular, certain postulated states/conformations of the remodeling of RAC on the ribosome upon engagement of the nascent chain as well as during hand-over of the nascent chain from RAC to Ssb necessitate biochemical or genetic verification. For example, postulated close proximity of certain residues of Zuo1 and Ssz1 or Zuo1 and Ssb could be probed by chemical cross-linking.

We fully understand this reviewer's concern. Several labs (Craig, Sinning and Rospert) have been focusing on the mechanism of the RAC-Ssb system using biochemical and genetic methods, including chemical cross-linking and non-natural amino-acid based system. Over years, they have achieved many important results which could be well aligned with our structural data. We do not have all these materials (various yeast mutant strains, a collection of antibodies, and the Bpa-TAG system in the case of *in vivo* crosslinking) to set up cross-linking experiments in a short time. Therefore, we wish to integrate our structural findings with these published biochemical data to advance our knowledge on the function of the RAC-Ssb system.

In the revision, we have presented a set of the new structures from another cryo-EM dataset (RNC-RAC without supplementing Ssb-ATP). One of these new structures have solid density for the full-length of Zuo1 and Ssz1 (Fig. 4). We believe that this is a major improvement over the initial submission, and the colleagues in the co-translational folding field would love to see this new structure.

- The old nomenclature is used for ribosomal proteins. The new nomenclature for ribosomal proteins (Ban et al., 2014 Curr. Opin. Struct. Biol.) should be used throughout the manuscript.

Suggestion is well taken.

- Some experimental details, e.g. the different types of stalling constructs and their description are too extensive in the main text and should be shortened. It would be easier to focus on the PPP stalling construct that was used for generation of the stalled RNC.

Initially, we have introduced the experiments in details to help the readers understand our experimental design. This part of strain construction has been shortened in the revision.

- What is the experimental proof for the blocking of the Zuo1 HPD motif by Ssz1 SBD? This has to be shown experimentally by mutations and crosslinking experiments.

Firstly, in the structure of the endogenous 80S-RAC complex, the HPD motif is covered by Ssz1 and not exposed (Supplementary Fig. 2, Map 1.4). Secondly, in our new structures from the RNC-RAC dataset (without additional Ssb) we have captured three different states of RAC (Supplementary Fig. 9). The C1 state (Fig. 5a) is similar to the endogenous 80S-RAC complex and the HPD motifs in the other two states (Fig. 5b, c) are still not compatible with the binding of Ssb-ATP (structural alignment based on highly conserved HSP40:HSP70 interface). We believe that these structural observations and analyses are reliable. Therefore, our conclusion that RAC undergoes structural remodeling to accommodate Ssb-ATP is convincing. We wish the reviewer to understand that it would be a challenge for us to provide results of crosslinking experiments in a timely manner.

- Can the postulated initial docking of Ssb on the RNC-RAC complex be verified experimentally (e.g. by chemical cross-linking)?

Based on the structural docking, the NBD of Ssb-ATP is distant from the ribosome, while the SBD is close to the PTE. And uL24 is the closest protein from the ribosome. In fact, crosslinking between uL24 and Ssb-SBD (ADP state) was very recently reported by Craig group (Lee et al., 2021). In principle, cross-linking between Zuo1 and Ssb in the conserved HSP40:HSP70 interface could also be probed by site-directed *in vivo* cross-linking experiments.

- What could be the function of the previously identified positively charged ribosome-binding region at the C-terminus of Ssb (in SBD α) in the model since in the “proposed” structure the Ssb C-terminus does not seem to contact the ribosome?

In our model, we superimposed the structure of Ssb-ATP onto the RNC-RAC complex (Fig. 6a). Upon ATP hydrolysis, HSP70 proteins including Ssb would undergo a substantial conformational change: SBD α would fold back onto SBD β to clamp the

substrate and the orientation between the NBD and SBD is also greatly altered. Therefore, it is completely reasonable that in the aligned Ssb-ATP structure the SBD α is free of ribosomal contact. We believe that after substrate binding and ATP hydrolysis, Ssb in its ADP-bound form should allow the repositioning of SBD α close to the PTE of the ribosome. This is exactly reported in the recent paper from Craig group (Lee et al., 2021), where an *in vivo* crosslink between uL24 and Ssb-SBD β (ADP conformation) was found.

- The reviewer could not locate Supplementary Table 1

Supplementary Table 1 was submitted in the form of excel format, which includes multiple sheets of all the results of mass spectrometry.

- Supplementary Figure 1A: According to the silver stained gel, it seems that a detectable amount of Ssb is also co-purified upon purification of tagged Zuo1 and ribosomes (also present in the pellet fraction indicating ribosome association). Did the authors detect Ssb in their 3D classification of the 80S-RAC particles?

The amount of Ssb is extremely low relative to other components and we did not observe any density of Ssb in the structures from the endogenous 80S-RAC dataset during 3D classification.

Minor points:

- line 56: typo “addition” should be “additional”
- line 290: “Supplementary Fig. 8c” should be “Supplementary Fig. 8b”

We thank the reviewer for catching these errors. They have been corrected.

Reviewer #3 (Remarks to the Author):

Structural insights into the function of ribosome associated Hsp40-Hsp70 complex in translation regulation by Chen et al. This manuscript investigates the molecular chaperone complex RAC and its cooperation with the Hsp70 Ssb during cotranslational protein folding. A particular goal is to describe the coordinated actions of RAC (Zuo1 and Ssz1) and Ssb1. To that end, the authors obtained cryo-EM structures of RAC bound to a nascent chain free ribosome and compared that to the structure of ribosome-nascent chain complexes. Overall they have a nice collection of different structures that contain different components of the translational machinery, and the structures provide more detailed information about how Zuo1 interacts with the ribosome. However, it isn't clear how the structures might progress from one complex to another and many of the steps in the model of Figure 6 are speculative, particularly when trying to discuss Ssb interaction and the presence of Stm1 and eIF5A

and eEF2 in their complexes. Overall, the main impact of this manuscript is a more detailed interaction about Zuo1, and it does not directly answer larger questions of handoff of newly translated protein from RAC to Ssb.

In the revised manuscript, we have added a new dataset for RNC-RAC complexes (without supplementing Ssb in sample preparation) and captured a set of new intermediate states. We have revised the model figure to focus on co-translational folding (Fig. 7). The text related to potential additional role of RAC and Stm1 was moved to the Discussion.

Specific comments

In general, the writing is not concise, and in many cases assumes that the reader is very familiar with previous structural and functional analysis of RAC. For example, “This suggests that Zuo1 present a physical and functional link between the PTE and the decoding center through its ES12-h44 interacting CTD (4HB)” (line 91).

We thank the reviewer for this suggestion. We have revised the text extensively. The text in line 91 has been rewritten.

It is difficult to interpret whether the RAC-80S structures are authentic functional steps considering that the 80S ribosomes are likely in a resting state (lines 138-142 and lines 166-169.)

We fully understand the reviewer’s concern. The RAC-80S structure is in a resting state. Since it does not contain a nascent polypeptide, we reason that it could be structurally equivalent to the initial binding of RAC on the ribosome. We have revised the text to avoid confusion.

In the revision, we have presented a set of new structures from another dataset (RNC-RAC without supplementing Ssb-ATP). The C1 state from this dataset is highly similar to the ones from the endogenous RAC-80S. Therefore, this new C1-type conformation is interpreted as the resting state in the revision.

The authors describe 3 different conformational states of RAC on the 60S that show conformational changes in Zuo1 and it is difficult to assess which is ‘correct’. One figure they focused on in Fig 1 Map 3.4 was based on only 9% of the particles. The RAC figures shown in Fig. 1, 2 and Supp 2 and 3 are somewhat poorly resolved and there is no obvious comparison to previously existing structures.

As we explained in the manuscript, RAC is extremely dynamic on the ribosome. Therefore, we have to rely on deep 3D classification to sort out a very small fraction of particles for structural refinement. In these refined structures, the ribosome parts could reach better than 3-Å resolution. Although the RAC regions in these structures are in 4 to 8-Å range, they are sufficiently resolved to fit in secondary structures of Zuo1 and

Ssz1. Therefore, the interactions among the RAC-Ssb components and between them and the ribosome could now be understood at least at residue level. We believe that this is already an important progress in the field.

We used Map 3.4 (Map 1.7 in the revised Supplementary Fig. 2) in Fig 1 in order to show the overview of the endogenous 80S-RAC complex. One reason is that Map 3.4 has the most complete factors. We could show a collection of maps in the figure, but it would be very crowded and too distracting for the readers.

Maps used in Fig. 1, 2 and Supplementary Fig. 2 and 3 were filtered according to the global resolution. RAC is located on the edge of the complex in relatively less resolved regions. There are only two previously reported low-resolution structures of 80S-RAC complex (Leidig et al., 2013; Zhang et al., 2014). Our new structures are consistent with them in the general appearance and the ribosomal location, but with much more details.

Supplementary Table 1 needs more description, such as details on which protein hits were significant and which ones were not.

We have revised Supplementary Table 1. The components identified in the structures are now highlight in red. The previous table contains a column of “Score” or “Sum PEP Score”. We wish to present the table in its raw form to provide more information for future studies.

There is also minimal validation of newly described structural interactions and that is limited to mutational analysis of Zuo1 and effect on growth. They make mutations in Zuo1 residues critical for ribosome interaction and show that they disrupt function *in vivo* but do not show that the mutations also disrupt ribosome interaction.

We designed mutation experiments to test whether these sites of interaction are critical for the *in vivo* function of RAC. The reason we did not test their ability in ribosome binding is that: (1) Many sites have already been tested in previous studies, for example, the basic residues in the 4HB region; (2) RAC interacts with the ribosome through multiple contact sites and involves a large number of residues. Single mutations normally could not give an enough contrast in the ribosome binding; (3) The results from ribosome binding and plasmid based growth rescue experiments often did not correlate with each other (Lee et al., 2016). Therefore, we think that it is more important to look at the fitness of these mutant.

The native substrate used for the RNC-RAC structure has a lot of modifications, raising questions about whether it is really still a native substrate after adding translational stalling motifs and eGFP?

We chose the N-terminal sequence of Pmt1 as the nascent chain because a previous

study of Ssb-nascent chain interactions revealed that Pmt1 had a very sharp Ssb-binding peak in its N-terminal sequence (Doring et al., 2017). Notably, this sequence is also among the most enriched Ssb-recognition sites (about 14-fold) in the cell.

We understand this concern. But it is sometimes a compromise one would have to take for structural studies. In our constructs, eGFP is used for the detection of stalling efficiency. Based on our design, eGFP coding sequence is downstream of the stalling motif and therefore should not be translated.

Fig 2 is a composite map of a representative RNC-RAC-Ssb1 complex built from the the 40S subunit from Map 5.1 and the 60S subunit plus tRNAs from Map 4.2, not an intact structure, and Ssb was further modeled onto that structure. Again, given the heterogeneity of the samples, it is difficult to assess whether these are authentic structures.

The use of composite maps is that we would like to show the features of the structure in a single panel. Because the NTD and CTD of Zuo1 is connected by an α -helix. We have to refine the two parts separately in two reconstructions. For example, Fig 2b and 2c (now supplementary Fig. 8) were prepared with globally refined maps, in which the density of RAC is very weak due to the conformational averaging and only start to appear in much lower contour levels. Therefore, we used composite maps only to give very brief and overall structural information. Also, to avoid confusion, we have stated this information in the figure legend in the previous submission. In addition, we have also clearly indicated the versions of maps used for figure preparation in figure legends, which we believe is a must for cryo-EM structural presentation but not followed by many other groups.

Reviewer #4 (Remarks to the Author):

This work focuses on the structural basis of 80S ribosome-binding by the Ribosome Associated co-chaperone complex (RAC). The authors purified and reconstructed multiple instances of budding yeast RAC-ribosome complexes, some of which reached sufficiently fine resolutions to model unknown aspects of the chaperone's structures and ribosome recognition motifs. In addition, by exploiting different tagging and purification strategies, the authors made surprising observations about 80S-RAC complexes with, and without, nascent polypeptide chains, tRNAs, and other translation-associated factors.

First, by tagging Zuo1, one of the main components of RAC, the authors semi-purified endogenous 80S-RAC complexes. Surprisingly, the ribosomes reconstructed from this sample lacked mRNA, tRNA, or nascent chains. Rather, the predominant species was an entirely unexpected 80S-RAC-eEF2-eIF5A-Stm1 particle. The authors speculate about the meaning of this particle--invoking potential roles for RAC in termination or

hibernation--but without clear functional insights into the roles of the co-chaperone on ribosomes that lack nascent polypeptides.

Second, the authors then designed over-expression constructs with elongation-stalling properties and expressed these in Dom34 knockout yeast cells. Under these conditions, stalled ribosomes with nascent chains accumulate to high levels with bound RAC. The authors purified and reconstructed these complexes (80S-RNC-RCA) to identify different locations and conformations of Zuo1's domains relative to the exit site of the protein translocation tunnel. This set of structures suggest how Zuo1 flexibly engages the ribosome using multiple, adjacent, and semi-stable poses. By integrating prior genetic and structural studies, the authors propose how Zuo1 may recruit and regulate Ssb's substrate engagement and ATP hydrolysis cycle.

Together, this paper includes multiple observations of interest, including the possibility that RAC has a role with inactive Stm1-blocked ribosomes. Before publication, we recommend addressing the following issues:

Major points:

1. We would like to suggest the authors consider separating the experimental results and more speculative aspects of the discussion. Many statements in this manuscript are deduced from the references or are speculation. In this case, it seems important to guide the reader through the observations and save the more daring speculations for later. For example, is it possible the unexpected nature of the particle obtained by tagging and pulling down Zuo1 is due to the affinity tag on Zuo1? This seems as likely as the possibility the authors propose, that "...our endogenous 80S-RAC structure should resemble an initial state of RAC binding on the 80S prior to nascent chain engagement." Maybe, but with no tRNA, no mRNA, and no nascent chain?

In the revised manuscript we separated the experimental results and the discussion as suggested.

The 3xFLAG-tag on Zuo1 is at the C-terminus, which does not participate in the interaction with the ribosome and should have neglectable effect on the normal function of RAC.

We have changed our wording on this sentence to avoid confusion. What we wanted to express is that the conformation of RAC in the 80S-RAC complex could be structurally equivalent to or mimic the state of the initial binding of RAC to the ribosome.

2. There does not appear to be interpretable density for Ssb in the maps of the RNC-RCA-Ssb complexes--is Ssb present in the complex? Do the authors have a reconstruction of the RNC-RCA complex (without adding the exogenous Ssb)? It could be informative to compare the structures of RNC-RCA before and after adding Ssb.

We really appreciate the reviewer for this suggestion. In the past 6 months, a major experiment we have done was to characterize the structures of the RNC-RAC samples without the exogenous Ssb. The results were very positive. As shown in Fig. 4 and Fig. 5, we have captured a set of new structures. One structure contains solid density for nearly full-length of Zuo1 and Ssz1.

Notably, in these new structures of the RNC-RAC dataset (without supplementing Ssb), the JD of Zuo1 is still not ready for Ssb-ATP accommodation. This indicates that upon adding exogenous Ssb, Ssz1 could be completely mobilized to fully expose the HPD motif.

3. Was Stm1 identified in the mass spectrometry? I did not see it annotated in the supplementary material. It looks like Stm1, but was the identification strictly from the cryoEM density?

Stm1 was identified in the mass spectrometry as listed in Table1 (sheet R2-R3, Accession P39015). In fact, the cryo-EM density matched well with the model of Stm1 (PDB 4V8Y).

4. Line 386-387. The authors claimed, "Ssz1 should remain attached to Zuo1 through the flexible N-terminal sequence of Zuo1 NTD.". Are there any experimental data or references to support this statement?

RAC is extremely stable as a dimer through the tight interaction between the N-terminal sequence of Zuo1 and Ssz1. Based on the crystal structure of Ssz1-ND_{Zuo1} (Weyer et al., 2017) the N-terminal 55 residues of the ND have extensive interactions with the SBD of Ssz1 (also with the NBD) (Rebuttal Figure 4). The first helix of the ND (residues 44-55 in *C. thermophilum* numbering, 34-45 in *S. cerevisiae* numbering) wraps around a linker helix between the NBD and SBD of Ssz1. Therefore, for Zuo1 to completely dissociate from Ssz1, the ND sequence has to be threaded from the cavity formed between the SBD and NBD of Ssz1. This would be very inefficient.

In fact, in all the published purification experiments, Zuo1 and Ssz1 always co-exist in same cellular fractions and display a 1:1 molecular ratio, indicating that they are likely a obligate dimer in the cytoplasm. Rospert and Sinning groups showed that the nascent chain could compete with Zuo1-ND for the binding site on Ssz1-NBD (Zhang et al., 2020). This replacement of Zuo1-ND with the nascent chain takes place from the N-terminus of Zuo1-ND (Fig. 6), and is likely unable to strip all the Zuo1-ND sequences from Ssz1-BD (Rebuttal Figure 4).

Rebuttal Figure 4. The crystal structure of *C. Thermophilum* Ssz1-ND_{Zuo1} complex (adapted from Weyer et al., 2017).

5. Atomic model building and validation. The models were primarily built from existing structures with rigid-body docking, although manual tuning in Coot is mentioned in the methods. Please provide standard validation metrics, for example from MOLPROBITY, or Phenix and structural statistics table.

Supplementary Table 2 has been updated.

Minor points:

1. Typographic errors on line 281 “displays both translational and rotational of movement relative to the 60S” and on line 339. “... several single lysine (K341A, K342, 344A and K348) or ...” should be “... several single lysine (K341A, K342A, K344A and K348A) or ...”

We thank the reviewer for catching the errors. They have been corrected.

2. Fig. 5b. Please label the HPD motif in the figure.

Fig. 5b in the last version was removed in the revision.

3. Supplementary Figure 11c. Is it Ssb1-ADP or Ssb1-apo?

Supplementary Fig. 11c in the last version was removed. It is probably a part of Ssb1-ADP after the ATP hydrolysis. However, since we could not exclude other possibilities, we decided to remove this discussion in the revision

4. Please make sure that the EM density maps and models are uploaded to EMDB and PDB. Also please provide the PDB/EMDB numbers in the manuscripts.

The EM density maps and models were uploaded to EMDB and PDB.

5. The “substrate relay” model is mentioned in the abstract and cited in the discussion, but never explained. What is this model and how does it pertain to this study?

The “substrate relay” model was proposed in the work of Sinning and Rospert labs (Zhang et al., 2020). Through the chemical-crosslinking experiments they proposed that the three components in the RAC-Ssb system interact with the emerging nascent chain in a sequential order. Zuo1 is the first and followed by Ssz1. And the nascent chain is finally handed over to Ssb by RAC. This relay process occurs during the elongation of nascent chain. Our structures show that RAC undergoes substantial and step-by-step structural remodeling upon substrate binding, which fits well with this substrate relay model.

6. The manuscript uses the term “in vivo” in many instances that are engineered, like exogenous over-expression of a stalling nascent chain in a dom34 null background. It seems in vivo does not convey a valid meaning for these conditions.

For most places, we used the term “*in vivo*” to refer to those cell-based results, rather than *in vitro* binding or *in vitro* translation methods. We have double checked these instances, and unnecessary use of this term is now removed.

References:

Doring, K., Ahmed, N., Riemer, T., Suresh, H.G., Vainshtein, Y., Habich, M., Riemer, J., Mayer, M.P., O’Brien, E.P., Kramer, G., *et al.* (2017). Profiling Ssb-Nascent Chain Interactions Reveals Principles of Hsp70-Assisted Folding. *Cell* 170, 298–311 e220.

Gumiero, A., Conz, C., Gese, G.V., Zhang, Y., Weyer, F.A., Lapouge, K., Kappes, J., von Plehwe, U., Schermann, G., Fitzke, E., *et al.* (2016). Interaction of the cotranslational Hsp70 Ssb with ribosomal proteins and rRNA depends on its lid domain. *Nat Commun* 7, 13563.

Hanebuth, M.A., Kityk, R., Fries, S.J., Jain, A., Kriel, A., Albanese, V., Frickey, T., Peter, C., Mayer, M.P., Frydman, J., *et al.* (2016). Multivalent contacts of the Hsp70 Ssb contribute to its architecture on ribosomes and nascent chain interaction. *Nat Commun* 7, 13695.

Kisonaite, M., Wild, K., Lapouge, K., Ruppert, T., and Sinning, I. (2022). High-resolution structures of a thermophilic eukaryotic 80S ribosome reveal atomistic details of translocation. *Nat Commun* 13, 476.

Lee, K., Sharma, R., Shrestha, O.K., Bingman, C.A., and Craig, E.A. (2016). Dual interaction of the Hsp70 J-protein cochaperone Zuo1 with the 40S and 60S ribosomal subunits. *Nat Struct Mol Biol* 23, 1003–1010.

Lee, K., Ziegelhoffer, T., Delewski, W., Berger, S.E., Sabat, G., and Craig, E.A.

(2021). Pathway of Hsp70 interactions at the ribosome. *Nature Communications* *12*.

Leidig, C., Bange, G., Kopp, J., Amlacher, S., Aravind, A., Wickles, S., Witte, G., Hurt, E., Beckmann, R., and Sinning, I. (2013). Structural characterization of a eukaryotic chaperone-ribosome-associated complex. *Nat Struct Mol Biol* *20*, 23–U34.

Strahl-Bolsinger, S., and Scheinost, A. (1999). Transmembrane topology of pmt1p, a member of an evolutionarily conserved family of protein O-mannosyltransferases. *J Biol Chem* *274*, 9068–9075.

Weyer, F.A., Gumiero, A., Gese, G.V., Lapouge, K., and Sinning, I. (2017). Structural insights into a unique Hsp70–Hsp40 interaction in the eukaryotic ribosome-associated complex. *Nat Struct Mol Biol* *24*, 144–151.

Zhang, Y., Ma, C., Yuan, Y., Zhu, J., Li, N., Chen, C., Wu, S., Yu, L., Lei, J., and Gao, N. (2014). Structural basis for interaction of a cotranslational chaperone with the eukaryotic ribosome. *Nat Struct Mol Biol* *21*, 1042–1046.

Zhang, Y., Valentin Gese, G., Conz, C., Lapouge, K., Kopp, J., Wolfle, T., Rospert, S., and Sinning, I. (2020). The ribosome-associated complex RAC serves in a relay that directs nascent chains to Ssb. *Nat Commun* *11*, 1504.

REVIEWERS' COMMENTS

Reviewer #1 (Remarks to the Author):

I read through the revised manuscript "Structural remodeling of ribosome associated Hsp40-Hsp70 chaperones during co-translational folding" by Chen et al. I find the issues raised by myself and also the other referees well addressed and the manuscript much improved. The work in my view is ready for publication. It is an important piece of work.

One tiny suggestion (entirely optional): How about using the specific names of the Hsp40/70 components (RAC/Ssb, or Zuo1/Ssz1/Ssb) in the title? This occurs to me, because I find the earlier work from 2014 (Structural basis for interaction of a cotranslational chaperone with the eukaryotic ribosome) pretty hard to find in literature searches.

Reviewer #2 (Remarks to the Author):

The revised version of the manuscript by Chen et al. has significantly improved. Importantly, a new dataset (RNC-RAC) that has been added in the revised version adds a depth of new information on the states of Zuo1 and Ssz1 on the ribosome. In this context, contact points (some of which known before as well new ones) of Zuo1 and Ssz1 with the ribosome could be visualized at high resolution. For one state almost full-length Zuo1 and Ssz1 (RAC) could be resolved on the ribosome (representing the first almost complete structure of RAC on the ribosome). In addition, the different states (from RNC-RAC and RNC-RAC-Ssb datasets) allow to propose a plausible and intriguing model of RAC activation and movement on the ribosome. Overall, the presented structural work represents a valuable contribution to the field and clearly extends our understanding of the functioning of the RAC complex.

In our first review we commented that several findings require additional biochemical and/or genetic verification. The authors did not add any new biochemical/genetic data and explain this by the fact that they don't have the technical means and expertise to conduct such experiments (within a limited period of time). Although we still think that such an experimental verification of some of the author's findings would significantly strengthen the study, we in the light of the new abovementioned structures, feel no longer that additional experiments are required to warrant publication.

However, there are still a number of issues that need to be addressed/improved before publication:

- While figures, outline and the text of the manuscript have been improved, the manuscript is still difficult to read (in certain parts) and the text still requires extensive revision (both with respect to grammar and spelling). We suggest to include editing by a native speaker.

- As suggested by the Reviewer, in the results section the Smt1 part (and the part discussing 80S-RAC) has been shortened appropriately. However, in the reviewer's opinion, the Stm1 part in the discussion section still needs revision (some sentences don't make sense).

Questions/Comments

- According to the proposed model, Ssz1 needs to dissociate from Zuo1 to allow for Ssb-ATP binding to Zuo1 on the ribosome. Zuo1-NTD and Ssz1 bind tightly. No data so far indicate that the RAC complex would dissociate rather quite the opposite; RAC heterodimer is known to be very stable. How do the authors envisage dissociation of Ssz1 from Zuo1? This has to be discussed with caution and very critically.

- In a recent report from the Craig lab, Lee et al., 2021, a Ssz1/Ssb heterodimer has been postulated. Does (and if so at which stage) such a heterodimer fit into the proposed model?

- What is the explanation for the fact that no densities for either Ssz1 or Ssb1 can be seen in the RNC-RAC-Ssb dataset?

- comment: In general, the generation of complexes for cryo EM analysis (RNC-RAC or RNC-RAC-Ssb; i.e. purification and supplementation) seems rather artificial and thus the physiological relevance of these data is still to be tested. This needs to be clearly stated.

Reviewer #3 (Remarks to the Author):

The revised manuscript provides a cohesive description of how Rac and Ssb interact with the ribosome. It is very much improved and this reviewer has no additional suggestions.

Reviewer #5 (Remarks to the Author):

In the revised manuscript, the authors have answered our questions and provided exciting new data to further our understanding. The manuscript should be published without delay.

First, the authors determined another reconstruction of the RNC-RAC complex without adding Ssb1. This new map fills in missing aspects of their model for the sequential steps of co-translational protein folding. Second, the manuscript has been extensively revised to cleanly separate results and discussions. Lastly, they have revised the discussion to provide a scholarly overview of potential functions of Stm1, eIF5A, and eEF2 in these complexes. The findings are indeed very exciting and important to the field, and demonstrate the power of multi-class cryo-EM analyses for inferring mechanisms.

We have just two questions/suggestions to consider during the final publication process:

1. The rebuttal letter and original submission included a reasonable explanation of how the authors are thinking about the nucleotide-state of Ssb1 (e.g. original Fig. 5). We found these speculations helpful and hope the authors will consider including them in the final version.
2. In the paragraph 'Mechanism of RAC in promoting the chaperone function of Ssb'. We are still of the view that the RNC-RAC-Ssb map does not have clear enough density to support some of the statements in the manuscripts (for example: Line 350-352, Line 354-361, and Line 340-344.) Please consider softening these conclusions, but this should not preclude or delay publication.

Minor suggestions:

1. Line 409. This is the first time the authors mentioned the LP motif, please label it in the figures.
2. Typographic error in Fig. 1 legends. Line 954: peptidyl transferase center should be 'PTC', not 'PTE'.
3. Line 290 and Fig 4c. Could the authors specifically indicate which residues (or regions) in SBD and Zuo1 participate in the polar interaction? The current descriptions are a bit vague.
4. Line 408-409. '...Zuo1-ND on Ssz1-SBD...' we found this confusing. Do the authors mean that it is '...Zuo1-ND that is interacting with Ssz-SBD...' or '...Zuo1-ND binding with Ssz-SBD...' ?

REVIEWERS' COMMENTS

Reviewer #1 (Remarks to the Author):

I read through the revised manuscript "Structural remodeling of ribosome associated Hsp40-Hsp70 chaperones during co-translational folding" by Chen et al. I find the issues raised by myself and also the other referees well addressed and the manuscript much improved. The work in my view is ready for publication. It is an important piece of work.

One tiny suggestion (entirely optional): How about using the specific names of the Hsp40/70 components (RAC/Ssb, or Zuo1/Ssz1/Ssb) in the title? This occurs to me, because I find the earlier work from 2014 (Structural basis for interaction of a cotranslational chaperone with the eukaryotic ribosome) pretty hard to find in literature searches.

Thanks for the suggestion. We wish to use a more general title, as the information provided here should be useful towards other eukaryotic RAC systems.

Reviewer #2 (Remarks to the Author):

The revised version of the manuscript by Chen et al. has significantly improved. Importantly, a new dataset (RNC-RAC) that has been added in the revised version adds a depth of new information on the states of Zuo1 and Ssz1 on the ribosome. In this context, contact points (some of which known before as well new ones) of Zuo1 and Ssz1 with the ribosome could be visualized at high resolution. For one state almost full-length Zuo1 and Ssz1 (RAC) could be resolved on the ribosome (representing the first almost complete structure of RAC on the ribosome). In addition, the different states (from RNC-RAC and RNC-RAC-Ssb datasets) allow to propose a plausible and intriguing model of RAC activation and movement on the ribosome. Overall, the presented structural work represents a valuable contribution to the field and clearly extends our understanding of the functioning of the RAC complex.

In our first review we commented that several findings require additional biochemical and/or genetic verification. The authors did not add any new biochemical/genetic data and explain this by the fact that they don't have the technical means and expertise to conduct such experiments (within a limited period of time). Although we still think that such an experimental verification of some of the author's findings would significantly strengthen the study, we in the light of the new abovementioned structures, feel no longer that additional experiments are required to warrant publication.

We sincerely thank the reviewer for his/her understanding of our technical difficulties.

However, there are still a number of issues that need to be addressed/improved before publication:

- While figures, outline and the text of the manuscript have been improved, the manuscript is still difficult to read (in certain parts) and the text still requires extensive revision (both with respect to grammar and spelling). We suggest to include editing by a native speaker.

Thanks. We have asked a colleague to help us improve the writing.

- As suggested by the Reviewer, in the results section the Smt1 part (and the part discussing 80S-RAC) has been shortened appropriately. However, in the reviewer's opinion, the Stm1 part in the discussion section still needs revision (some sentences don't make sense).

Suggestion is well taken. This part has been revised.

Questions/Comments

- According to the proposed model, Ssz1 needs to dissociate from Zuo1 to allow for Ssb-ATP binding to Zuo1 on the ribosome. Zuo1-NTD and Ssz1 bind tightly. No data

so far indicate that the RAC complex would dissociate rather quite the opposite; RAC heterodimer is known to be very stable. How do the authors envisage dissociation of Ssz1 from Zuo1? This has to be discussed with caution and very critically.

As we have discussed in the manuscript, we do not think a complete dissociation of Ssz1 from Zuo1 is involved in the Zuo1/Ssz1/Ssb1 functional cycle. The growing nascent chain only disrupts the interaction between the very N-terminal sequence of Zuo1 and Ssz1-SBD. This disruption would increase the flexibility of Ssz1 relative to Zuo1/ribosome such that Ssb could be accommodated.

We have further revised the text to avoid possible misunderstanding.

- In a recent report from the Craig lab, Lee et al., 2021, a Ssz1/Ssb heterodimer has been postulated. Does (and if so at which stage) such a heterodimer fit into the proposed model?

The Ssz1/Ssb heterodimer proposed in the recent paper by Lee et al. is consistent with our model. They have found that Ssb in the ATP-binding state displays specific crosslinks with Ssz1, which implies that upon the initial docking of Ssb-ATP, Ssz1 could have direct contact with Ssb1. Based on our structural information of the RNC-RAC dataset, Ssz1 could sample multiple sharply different orientations (Figure 5) before the binding of Ssb-ATP. A possible Ssz1/Ssb heterodimer is not in conflict with our model. Since we do not have the structural information for the RNC-RAC-Ssb (ATP state), we did not include this discussion in the manuscript.

- What is the explanation for the fact that no densities for either Ssz1 or Ssb1 can be seen in the RNC-RAC-Ssb dataset?

Because we have included ATP in the sample preparation, the dataset of RNC-RAC-Ssb would presumably reflect the functional state of Ssb that is after ATP-hydrolysis and formation of the stable substrate interaction. In this condition, Ssz1 have already been mobilized by the nascent chain and Ssb-ADP could have lost its direct ribosomal association as well.

We have included some of these discussions in the revision.

- comment: In general, the generation of complexes for cryo EM analysis (RNC-RAC or RNC-RAC-Ssb; i.e. purification and supplementation) seems rather artificial and thus the physiological relevance of these data is still to be tested. This needs to be clearly stated.

We thank the reviewer for this suggestion. We have checked the manuscript and added a statement in the first paragraph of the Discussion.

Reviewer #3 (Remarks to the Author):

The revised manuscript provides a cohesive description of how Rac and Ssb interact with the ribosome. It is very much improved and this reviewer has no additional suggestions.

Reviewer #5 (Remarks to the Author):

In the revised manuscript, the authors have answered our questions and provided exciting new data to further our understanding. The manuscript should be published without delay.

First, the authors determined another reconstruction of the RNC-RAC complex without adding Ssb1. This new map fills in missing aspects of their model for the sequential steps of co-translational protein folding. Second, the manuscript has been extensively revised to cleanly separate results and discussions. Lastly, they have revised the discussion to provide a scholarly overview of potential functions of Stm1, eIF5A, and eEF2 in these complexes. The findings are indeed very exciting and important to the field, and demonstrate the power of multi-class cryo-EM analyses for inferring mechanisms.

We have just two questions/suggestions to consider during the final publication

process:

1. The rebuttal letter and original submission included a reasonable explanation of how the authors are thinking about the nucleotide-state of Ssb1 (e.g. original Fig. 5). We found these speculations helpful and hope the authors will consider including them in the final version.

Thank you. We have revised the manuscript to discuss the possible nucleotide-state of Ssb1, with regard to the explanation of the RNC-RAC-Ssb1 dataset.

2. In the paragraph 'Mechanism of RAC in promoting the chaperone function of Ssb'. We are still of the view that the RNC-RAC-Ssb map does not have clear enough density to support some of the statements in the manuscripts (for example: Line 350-352, Line 354-361, and Line 340-344.) Please consider softening these conclusions, but this should not preclude or delay publication.

The suggestion is taken. We have softened our tones in this section.

Minor suggestions:

1. Line 409. This is the first time the authors mentioned the LP motif, please label it in the figures.

The suggestion is taken. We have labeled the LP motif in Supplementary Fig. 12, which is in the very N-terminus of Zuo1. We also tried to label it in Fig. 4c. but it did not display well.

2. Typographic error in Fig. 1 legends. Line 954: peptidyl transferase center should be 'PTC', not 'PTE'.

We thank the reviewer for catching this error. It has been corrected.

3. Line 290 and Fig 4c. Could the authors specifically indicate which residues (or regions) in SBD and Zuo1 participate in the polar interaction? The current descriptions

are a bit vague.

This polar interface involves residues E83-D89 of Zuo1, and R521, K534 from Ssz1-SBD. We have also added this information in the manuscript.

4. Line 408-409. ‘...Zuo1-ND on Ssz1-SBD...’ we found this confusing. Do the authors mean that it is ‘...Zuo1-ND that is interacting with Ssz-SBD...’ or ‘...Zuo1-ND binding with Ssz-SBD...’?

We thank the reviewer for this correction. It should mean “Zuo1-ND that is interacting with Ssz1-SBD”. We have corrected it in the manuscript.